# Neural circuit for social authentication in song learning

**Jelena Katic** [1], **Yuichi Morohashi** [1] **& Yoko Yazaki-Sugiyama** [1,2] ✉

Social interactions are essential when learning to communicate. In human speech and bird song, infants must acquire accurate vocalization patterns and learn to associate them with live tutors and not mimetic sources. However, the neural mechanism of social reality during vocal learning remains unknown. Here, we characterize a neural circuit for social authentication in support of accurate song learning in the zebra finch. We recorded neural activity in the attention/arousal state control center, the locus coeruleus (LC), of juvenile birds during song learning from a live adult tutor. LC activity increased with real, not artificial, social information during learning that enhanced the precision and robustness of the learned song. During live social song learning, LC activity regulated long-term song-selective neural responsiveness in an auditory memory region, the caudomedial nidopallium (NCM). In accord, optogenetic inhibition of LC presynaptic signaling in the NCM reduced NCM neuronal responsiveness to live tutor singing and impaired song learning. These results demonstrate that the LC-NCM neural circuit integrates sensory evidence of real social interactions, distinct from song acoustic features, to authenticate song learning. The findings suggest a general mechanism for validating social information in brain development.

In vertebrate species with vocal communication, early auditory learning is more effective when acoustic training is accompanied by authentic social interactions with a live adult tutor. Vocal exposure with real social interactions in human infants triggers the development of phoneme detection[1], while passive auditory exposure is insufficient for successful speech development[2]. Likewise, juvenile songbirds learn to sing effectively through vocal communication with live tutors but develop poor song quality after passive exposure to recorded playback of tutor songs[3–5]. In juvenile zebra finches, song learning improves when they trigger song playback[6], suggesting that internal state, such as attention or motivation, enhances learning. Recent evidence suggests a role for neuromodulation in song learning. The midbrain periaqueductal gray (PAG) facilitates vocal copying for cultural transmission via dopamine signaling[5]. Another major neuromodulation command center for noradrenergic (NE) signaling, the locus coeruleus (LC), is known to modulate attention and arousal, broadcasting internal state information across the brain. LC neuronal activity modulates

behavioral processes such as long-term memory, sensory perception, and motivation to facilitate learning and memory[7–11]. Exposure of juvenile birds to a live, singing tutor increases the expression of immediate early genes in LC neurons compared to control juveniles passively exposed to the same songs through a speaker[4]. LC neurons anatomically project to avian higher auditory cortex, the caudomedial nidopallium (NCM)[12], a proposed brain locus for memory formation of the tutor song[13,14]. We recently reported that a subset of NCM neurons selectively respond to tutor song, and that the auditory responses of those neurons increase in the presence of a tutor[13,15]. However, whether NCM neural activity can integrate social information from a tutor, apart from acoustic features of tutor song, via the LC remains unknown. In this study, we recorded and manipulated neuronal activity in the LC-NCM neural circuit of juvenile zebra finches learning to sing from a socially-interacting tutor. We observed enhanced neural activity in the LC-NCM neuronal circuit during vocal communication with a live adult tutor. Juveniles, in which the LC-NCM neural circuit

---

[1]Neuronal Mechanism for Critical Period Unit, Okinawa Institute of Science and Technology (OIST) Graduate University, Okinawa, Japan. [2]WPI-IRCN, The University of Tokyo, Tokyo, Japan. ✉e-mail: yazaki-sugiyama@oist.jp

was optogenetically inhibited during exposure to a live singing tutor, did not learn the tutor's song, suggesting that this neuromodulatory mechanism integrates and authenticates social information concurrent with the processing of prosodic patterns for accurate song learning.

## Results

### Social interactions increase neuronal activity in the LC and NCM

Juvenile zebra finches tutored by a live, socially interacting male adult develop more precise and robust song learning compared to those passively exposed to tutor songs, and they show immediate early gene expression in the LC[4], which is reported to control attention levels in mammals[8–10]. Here we recorded single neuron activity in the LC or NCM of freely moving juvenile zebra finches when they were alone or socially interacting with a tutor to see whether the LC-NCM neural circuit encodes information of social interactions with a tutor to enable song learning. We recorded 352 neurons from either the LC or NCM over the period of tutoring (Supplementary Table 1). The number of recorded neurons was relatively smaller than previous studies in rodents[7,9,16,17] or studies that used anesthetized birds[12,18,19]. However, electrophysiological neuronal recordings in freely moving juvenile birds are restricted because of their body size and standing posture. Therefore, our data was comparable to studies in which single unit activity was recorded in freely moving songbirds[5,13]. Birds were presented with ~20 min of passive song playback of each of four different song stimuli: a future tutor song (TUT), two conspecific adult zebra finch songs (CON1 and CON2), and one heterospecific song of a Bengalese finch bird (HET). The playback session (Playback 1) was followed by 60–120 min of exposure to a live, singing tutor (LIVE TUT) with a 30 min-long interval, then another 20 min of passive playback (Playback 2) with a 30 min-long interval again (Fig. 1a). Recent evidence in rodents reported a heterogenous LC neuronal population comprised of one noradrenergic and two GABAergic cell types based on their waveforms and modes of activation[16], therefore, we classified 29 recorded LC neurons from 12 juveniles based on their firing rate as either regular-spiking or fast-spiking based on their firing rates (Supplementary Fig. 1a–c). Fast-spiking neurons showed similar spike durations and shapes (Supplementary Fig. 1a left, Supplementary Fig. 1b). In contrast, regular-spiking neurons exhibited greater variation in both spike shape and duration (Supplementary Fig. 1a right, Supplementary Fig. 1b), suggesting that regular spiking neurons are composed of heterogeneous neuronal subtypes as previously suggested in rodents[16]. To see the effect of social interaction with a tutor on song learning clearly, we focused on neuronal activities recorded in the first day of tutoring when we can compare the effect of exposure to live tutor singing without compromising the effects of tutoring of previous days. Thirteen out of 29 neurons were lost during the repeated song presentation or tutor song exposure which lasted for some hours, so that the remaining 16 neurons (7 fast- and 9 regular-spiking), which were recorded in the first day of tutoring, were further analyzed for song responsiveness. Interestingly, all the fast-spiking LC neurons paused their firing for a moment ($1.62 \pm 0.25$ s, $n = 7$) in response to the introduction of the tutor in the cage, but no regular-spiking neurons stopped their firing (Supplementary Fig. 1d). Both regular- and fast-spiking LC neurons increased their firing in response to passive song playback (Supplementary Fig. 1e). The increase in firing was sustained throughout the song playback and the pattern of LC neuron activity varied across repeated presentations of the same song, indicating that neuronal responses were not associated with specific song features or syllables. LC neuron responses to LIVE TUT singing with social interactions were greater compared to those with passive TUT playback (Fig. 1b, c). Notably, after long-term exposure to tutor singing (~60 min), LC neurons increased their responses to playbacks of all song stimuli, compared to responses before the exposure to LIVE TUT singing (Fig. 1 b–e). We found some LC neurons, which were recorded

later than the second day of tutoring, showed slightly but significantly greater responses to LIVE TUT than the TUT playback, but did not sustain higher responses after hearing the LIVE TUT (Supplementary Fig. 1f). However, we found none of the LC neurons developed selective auditory responses to TUT even after exposure to tutor singing on the first day of tutoring as $d'$ values, comparing responses to TUT and other songs, were between −0.5 to 0.5 (Fig. 1f), and response strength (RST) to any specific songs was not significantly different from that to other songs both before and after LIVE TUT exposure (Playback 1 and 2) (two-way ANOVA followed by Holm−Sidak post hoc test, $p > 0.05$) (Fig. 1d).

In contrast to LC activity, a subset of neurons in the NCM increased their RST specifically to TUT playback, but not to other songs after exposure to LIVE TUT (Supplementary Fig. 2a, b). Eighty NCM neurons were recorded from five juveniles in the first day of tutoring, and 57 of those were broad-spiking (BS) neurons according to their spike shape and firing rate[13]. We found that most of the BS neurons showed greater responses to LIVE TUT singing than to TUT song playback (Fig. 2a, Playback l vs LIVE TUT), but the RST to the TUT playback went back to a similar level after hearing LIVE TUT singing (Fig. 2a, Playback l vs 2, Supplementary Fig. 2d). However, we found a subset of BS NCM neurons ($n = 12$) showed greater responses to LIVE TUT singing while maintaining their song-aligned firing patterns (Fig. 2b, d, Playback l vs LIVE TUT) and kept their increased responses to TUT playback even after exposure to LIVE TUT (Fig. 2b, d, Playback 2). Those neurons responded selectively to TUT songs after hearing the LIVE TUT and did not increase the RST to playbacks of other songs (Fig. 2c, and Supplementary Fig. 2a, b). The RST to TUT was significantly higher than that to the playback of any other songs after hearing LIVE TUT (Playback 2), while it was not before the LIVE TUT (Playback 1) (two-way ANOVA followed by Holm−Sidak post hoc test, $p < 0.05$) (Supplementary Fig. 2b). The proportion of neurons showing selective responses to TUT playback, but not to other songs (TUT-selective neurons), significantly increased (from three to 12 neurons) after hearing the LIVE TUT (Fig. 2e, f). We further examined if the increase of RST to TUT song and/or the proportions of TUT selective neurons occurred in the later days of tutoring. We found a small number of neurons ($n = 4$) that increased their RST to TUT playback and began to show TUT-selective responses on the second day of tutoring. But later than the third day of tutoring, we did not find the neurons which increased RST or selectivity to TUT playback after hearing the LIVE TUT (Fig. 2f, g). In the third or fourth day of tutoring, some tutors did not sing songs at all for ~2 h of tutoring. In both cases, with or without LIVE TUT singing during social interactions with a tutor, we did not see the increase of RST to TUT song playback or proportions of TUT selective neurons (Fig. 2f, g). We found RST to TUT playback of TUT-selective BS neurons was significantly higher than that to the playback of other songs after the LIVE TUT (Playback 2), but not before (Playback 1) in the first day of tutoring. In contrast, RST to TUT was significantly higher than that to the playback of other songs both before and after the LIVE TUT later than the second day of tutoring regardless if a tutor sung (two-way ANOVA followed by Holm−Sidak post hoc test, $p < 0.05$) (Fig. 2g). We found a fraction of BS neurons that already showed selective responses to TUT before hearing LIVE TUT singing on the third and fourth day of tutoring (Fig. 2f). Those neurons did not show greater responses to LIVE TUT singing, neither did they increase their RST to TUT playback after being exposed to LIVE TUT (Fig. 2g and Supplementary Fig. 2c). We further found in the first day of tutoring that the narrower spiking (NS) NCM neurons responded greater but not significantly higher ($p = 0.1$) to LIVE TUT singing than to TUT song playback (Supplementary Fig. 2e). Some NS neurons showed an increased RST to TUT playback after hearing LIVE TUT singing, while none of them developed selective responses to TUT as they also increased their RST to other song stimuli (Supplementary Fig. 2f, g).

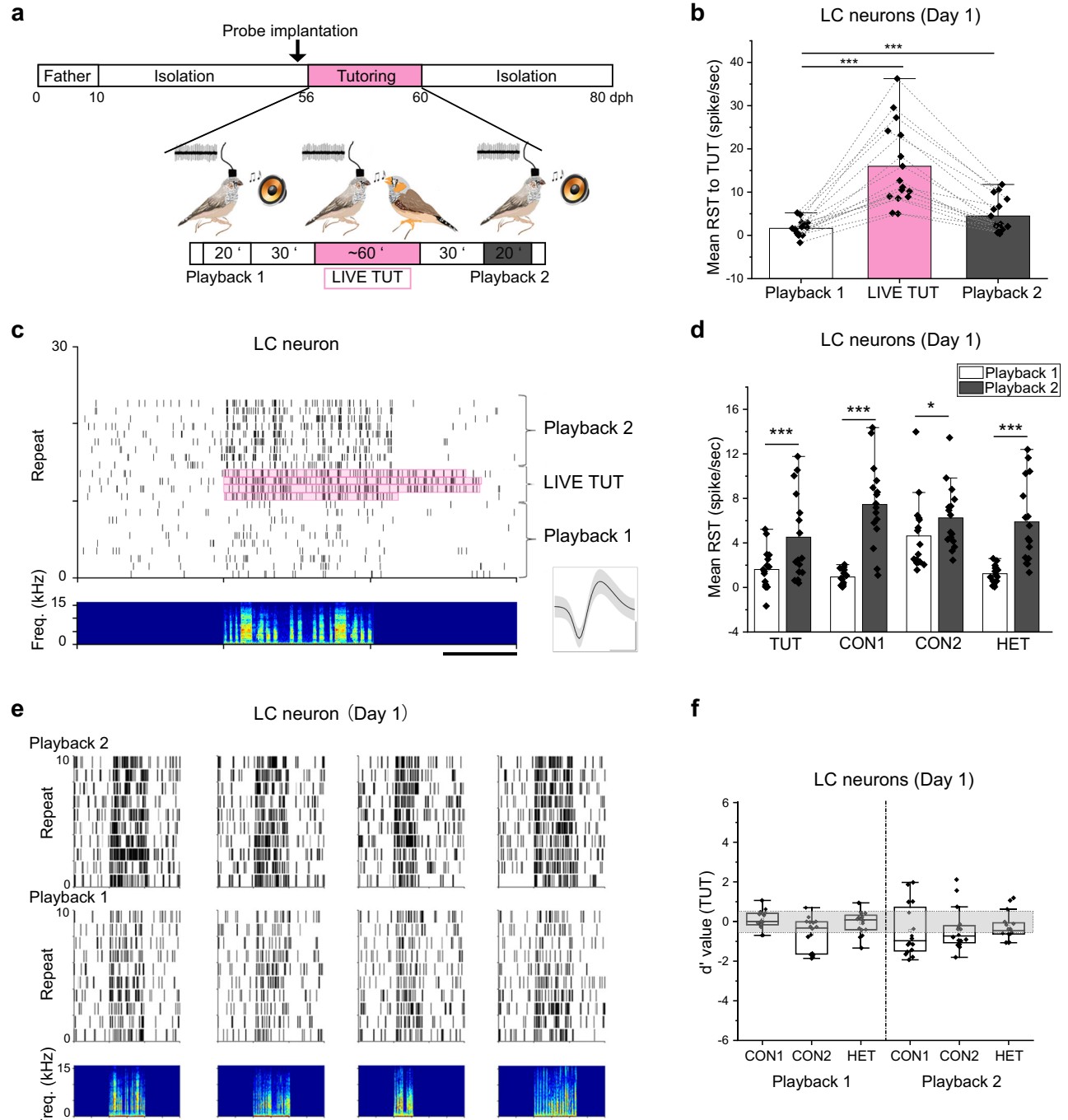

**Fig. 1 | LC neurons increase auditory responsiveness to the playback of all songs after social interaction with a singing tutor. a** Schematic drawings of the experimental design and timeline (dph = days, post hatch). Mean response strength (RST) of LC neurons to tutor song playback (Playback 1 and 2), tutor singing (LIVE TUT) (**b**) or playback of different songs before (Playback 1) and after (Playback 2) hearing LIVE TUT (**d**). Raster plots of spiking activities for a single LC neuron before, during and after LIVE TUT (**c**, scale bar: 1 s) or to playback of different songs before (playback 1) and after (playback 2) hearing LIVE TUT (**e**, scale bar: 2 s) (song spectrograms shown in the bottom). Inset: spike waveform of the same LC (**c**, mean ± s.d., scale bars: 0.5 ms horizontal, 0.5 mV vertical). **f** Mean d-prime values for TUT over other song stimuli of LC neurons before (Playback 1) and after (Playback 2) hearing LIVE TUT. The boxes show the 25–75%, the center lines are defined by the median and open squares by the mean. The whiskers include all data points within 1.5 IQR (Interquartile range) and the 'outsider' dots are the data points that fall outside the whisker line. Gray areas indicate non-selective responses (−0.5 < d' value < 0.5). $N = 8$, $n = 16$ (**b**, **d**, **f**). TUT: tutor song, CON1: conspecific song 1, CON2: conspecific song, HET: heterospecific song, $N$: number of birds, $n$: number of neurons. Mean ± s.e.m., *$p = 0.046$, ***$p < 0.001$ or $p = 0.0000251$ (HET), Two-sided Student T-test (HET, **d**) or Two-sided Mann−Whitney Rank Sum Test (**b**, **d**). Source data are provided as a Source Data file. The bird drawings in **a** were created by Nicolas Baudoin.

## Live tutor singing modulates NCM neuron activity via LC inputs

LC and NCM neurons in juveniles showed greater auditory responses to LIVE TUT with social interaction, and a fraction of the BS NCM neurons developed selective responses to TUT songs after exposure to LIVE TUT. LC neurons were reported to project anatomically to the NCM in adults[12,17]. We confirmed the LC-NCM projection in juveniles by injecting adeno-associated viral vectors (AAV2/9-hSyn-Cre, AAV2/9-FLEX-GFP) into the LC. We found GFP-positive axons in the NCM which

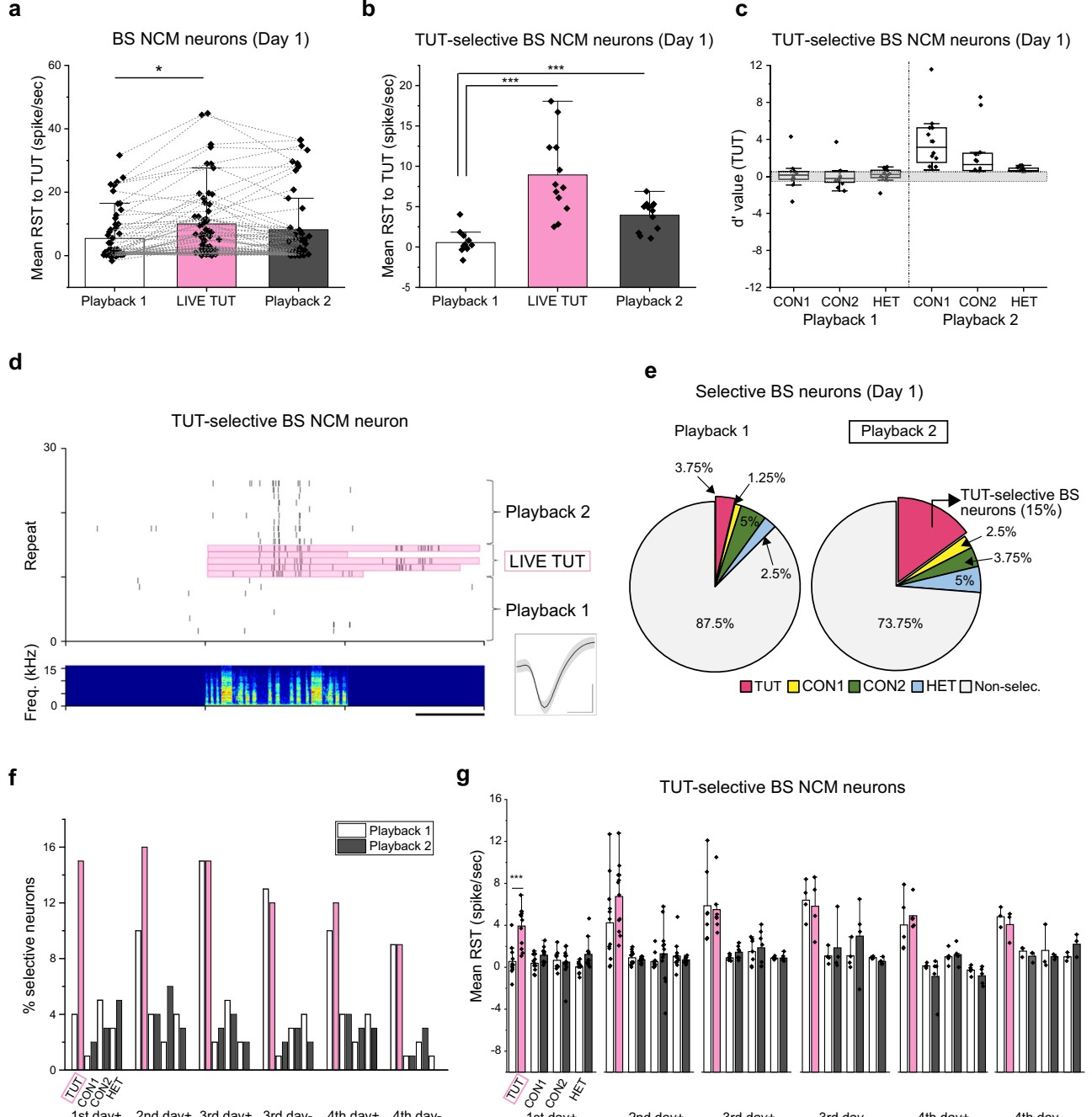

**Fig. 2 | NCM neurons increase auditory responsiveness to tutor song playback after social interaction with a singing tutor.** Mean response strength (RST) of all BS NCM neurons (**a**) or TUT selective BS neurons (**b**, **g**) to tutor song playback (Playback 1 and 2), tutor singing (LIVE TUT) (**a**, **b**) or playback of different songs before (Playback 1) and after (Playback 2) hearing LIVE TUT (+) or being exposed to a silent tutor (−) throughout four days of tutoring (**g**). **c** Mean d-prime values for TUT over other song stimuli of TUT selective BS neurons before (Playback 1) and after (Playback 2) hearing LIVE TUT. The boxes show the 25–75%, the center lines are defined by the median and open squares by the mean. The whiskers include all data points within 1.5 IQR (Interquartile range) and the 'outsider' dots are the data points that fall outside the whisker line. Gray areas indicate non-selective responses (−0.5 < d' value < 0.5). **d** Raster plots of spiking activity of TUT-selective BS NCM neuron to tutor song playback 1 and 2 or LIVE TUT (song spectrograms shown in the bottom) (scale bar: 1 s). Inset: spike waveform of the same TUT selective BS neuron (mean ± s.d., scale bars: 0.5 ms horizontal, 0.5 mV vertical). Proportion of BS NCM neurons that show selectivity to one song (**e**, **f**) or no selectivity (**e**, Non-selec.), before (Playback 1) and after (Playback 2) hearing LIVE TUT. $N = 5$, $n = 57$ (**a**), $n = 12$ (**b**, **c**), $n = 80$ (**e**, **f**: 1st day+), $n = 77$ (**f**: 2nd day+), $n = 44$ (**f**: 3rd day+), $n = 31$ (**f**: 3rd day−), $n = 44$ (**f**: 4th day+), $n = 30$ (**f**: 4th day−), $n = 12$ (**g**: 1st day+), $n = 12$ (**g**: 2nd day+), $n = 7$ (**g**: 3rd day+), $n = 4$ (**g**: 3rd day−), $n = 5$ (**g**: 4th day+), $n = 3$ (**g**: 4th day−). BS: broad-spiking neuron, TUT: tutor song, CON1: conspecific song 1, CON2: conspecific song, HET: heterospecific song, $N$: number of birds, $n$: number of neurons. Mean ± s.e.m., *$p = 0.0242$, ***$p = 0.0000143$, 0.0000518, Two-sided Student T-test (**a**, **b**, **g**). Source data are provided as a Source Data file.

were mostly immune-reactive with DBH or TH antibodies, but not with one to GABA (Supplementary Fig. 3a). We then investigated the degree to which LC neuronal activity modulated the auditory responsiveness of neurons in the NCM by optogenetically inactivating LC terminals in the NCM (Fig. 3a). Since we found that the NCM neurons develop TUT-selective responses in the first two days of tutoring, and that some tutors did not sing to juveniles after more than three days, we exposed juvenile birds to a tutor for three days with optogenetic stimulation.

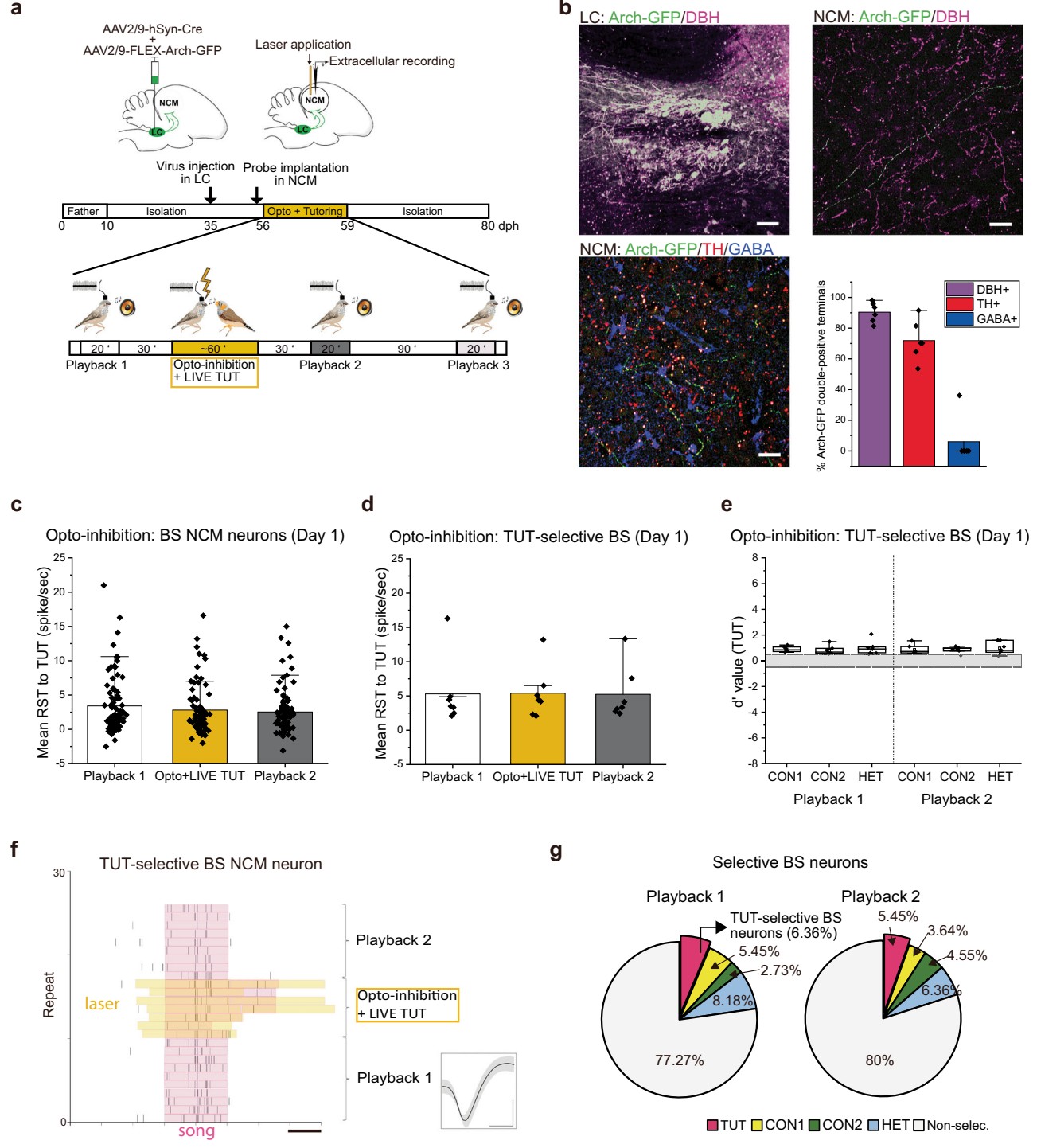

We injected a viral mixture (AAV-2/9-hSyn-Cre and AAV-2/9-FLEX-Arch-GFP) to express archaerhodopsin in the LC (Fig. 3b). Then, using an optoprobe (NeuroNexus, Buszaki16OCM16LP), we recorded the neuronal activity of NCM neurons while simultaneously inhibiting LC axons in the NCM, which were mostly DBH or TH positive (Fig. 3b), optogenetically only during the period when juveniles were hearing LIVE TUT, but not when they were merely interacting with a tutor (Fig. 3a). Most of the BS NCM neurons did not respond to TUT song playback before hearing LIVE TUT singing with social interaction in the first day of tutoring (Fig. 3c, Playback 1). They did not increase their response to LIVE TUT singing when LC inputs in the NCM were

optogenetically inhibited (Fig. 3c, OptoI+LIVE TUT), neither did they increase their RST to TUT playback after being exposed to LIVE TUT (Fig. 3c, Playback 2). A small subset of BS NCM neurons (*n* = 7), which exhibited selective auditory responsiveness to TUT playback before the exposure of LIVE TUT singing in the first day of tutoring, did not increase their responses to LIVE TUT and selectivity to TUT when optogenetic inhibition was applied during LIVE TUT (Fig. 3d–f). Optogenetic inhibition of LC axons in the NCM did not change the spontaneous firing rate of those neurons (2.63 ± 0.86 vs 2.31 ± 1.09, Playback 1 vs LIVE TUT+ Opto-inhibition). Moreover, the proportion of TUT selective BS neurons did not increase after being exposed to LIVE

**Fig. 3 | LC neuronal activity during social interaction with a singing tutor regulates NCM auditory responses and song selectivity to the tutor song.**
**a** Schematic drawings of the experimental design and timeline. **b** Top left: Parasagittal sections of the LC showing that LC neurons expressing Arch-GFP are dopamine beta-hydroxylase (DBH) positive (magenta). Parasagittal section in the NCM showing LC axonal terminals expressing Arch-GFP are DBH positive (magenta; top right) or tyrosine hydroxylase (TH) positive (red) but not GABA positive (blue; bottom left) (scale bars 100 um: top left, 20 um top right, and bottom left). Bottom right: Proportion of Arch-GFP axon terminals that are double-positive to DBH, TH or GABA. Mean response strength (RST) of all BS NCM (**c**) or TUT selective BS neurons (**d**) to tutor song playbacks (Playback 1 and 2) or tutor singing with optogenetic inactivation of LC inputs (Opto+LIVE TUT). **e** Mean d-prime values for TUT over other song stimuli of TUT selective BS neurons before (Playback 1) and after (Playback 2) hearing tutor singing with optogenetic inactivation of LC inputs. The boxes show the 25–75%, the center lines are defined by the median and open squares by the mean. The whiskers include all data points within 1.5 IQR (Interquartile range) and the 'outsider' dots are the data points that fall outside the whisker line. Gray areas indicate non-selective responses ($-0.5 < d'$ value $< 0.5$). **f** Raster plots of spiking activity in a TUT selective BS NCM neuron to tutor song playback 1 and 2 or Opto+LIVE TUT (scale bar: 1 s). Inset: spike waveform of the same TUT selective BS neuron (mean ± s.d., scale bars: 0.5 ms horizontal, 0.5 mV vertical). **g** Proportion of BS NCM neurons that show selectivity to one song or no selectivity (Non-selec.) before (Playback 1) and after (Playback 2) Opto+LIVE TUT. $N = 6$ (**b**–**e**, **g**), $n = 83$ (**c**), $n = 7$ (**d**, **e**), $n = 110$ (**g**). BS: broad-spiking neuron, TUT: tutor song, CON1: conspecific song 1, CON2: conspecific song, HET heterospecific song, N: number of birds, n: number of neurons. Mean ± s.e.m. **b**–**d** Two-sided Student T-test (**c**), Two-sided Mann–Whitney Rank Sum Test (**d**). Source data are provided as a Source Data file. The bird drawings in **a** were created by Nicolas Baudoin.

TUT singing coupled with the optogenetic inhibition of LC axons (Fig. 3g), in contrast to the control juvenile group (juveniles that expressed GFP in the LC and received the same laser application in NCM during LIVE TUT) in which both the proportion of tutor selective neurons and the RST to TUT playback increased after hearing LIVE TUT (Supplementary Fig. 3b, c). We found another subset of BS neurons ($n = 19$), which responded to TUT but also to another song playback (Fig. 4a, b, Playback 1) before being exposed to LIVE TUT singing in the first day of tutoring. They showed significantly diminished responses to LIVE TUT when the optogenetic inhibition was applied during LIVE TUT (Fig. 4a, b, Opto+LIVE TUT) and lost their responsiveness to TUT but not to other song playbacks at both 30 min (Fig. 4a, b and d, Playback 2) and 90 min (Fig. 4c, d, Playback 3) after being exposed to LIVE TUT singing with inactivation of LC. RST to TUT all before, 30 and 60 min after LIVE TUT singing was not significantly different from that to the other song playback (two-way ANOVA followed by Holm–Sidak post hoc test, $p > 0.05$) (Fig. 4d). Those results suggest that activation of LC inputs to the NCM by a live, singing tutor enabled NCM neuronal circuits to acquire TUT-selective auditory responses in specific neurons.

Over the period of three days of tutoring, the majority (76.4%, 84/110 neurons) of NCM neurons showed no significant change in their RST to TUT playback in juveniles where exposure to LIVE TUT was coupled with LC axon inactivation. Another 17.3 % ($n = 19$) of NCM neurons, predominantly BS neurons (68.4 %, $n = 13$), decreased their RST to TUT playback (Fig. 4e, Opto-inhibition, left). In contrast, in control juveniles that expressed GFP in the LC and were exposed to tutor singing coupled with laser application, about a half of NCM neurons (51.5%, 71/138 neurons) increased their RST to TUT playback, a half (45.1%, $n = 32$) of which was BS neurons (Fig. 4e, Control, right). While the number was small, a subset of otherwise silent NS neurons ($n = 9$) was activated only when LC terminals were optogenetically inhibited (aligned to the laser rather than to song onset, Supplementary Fig. 4a, b). These results suggest that vocal communication with a tutor is essential to modulate neuronal circuit activity in the NCM via functional inputs from the LC to acquire song-selective auditory responsiveness in the LC-NCM neural circuit.

**Song learning from live singing requires LC inputs in the NCM**
Our findings demonstrate that NCM neurons failed to develop selective auditory responses to TUT if LC-NCM inputs were inactivated during a juvenile's exposure to live tutor singing. Next, we examined if LC inputs to the NCM are necessary for juveniles to learn songs by socially interacting with a tutor. Juvenile birds that had undergone electrophysiological recordings with optogenetic inhibition (or laser application in the control group) were raised in isolation until they were adults. We then recorded their adult song and measured the similarity to their tutor's song (Fig. 5a)[20]. The songs of birds whose LC inputs to the NCM were inactivated during tutor singing with live social

interaction were significantly less similar to tutor songs compared to the songs of control birds (Fig. 5b, c). Further comparison of song similarity between adult zebra finch songs and other song stimuli to which they were exposed in the juvenile period, showed little similarity in either song in both control and LC-inactivated birds, indicating poor learning from song playback stimuli (Fig. 5c). Taken together, those results suggest that passive exposure to song playback did not lead to substantial learning while social exposure to a live singing bird only had an effect if the LC-NCM neural circuit was active during learning. To investigate if lesser similarities to tutor songs were caused by losing the ability to change vocal motor patterns with LC inactivation, we tracked the song similarity of juvenile songs to their final adult form. Control juvenile birds gradually increased their song similarity to their final adult song, indicating flexible learning of their song to their adult form (Fig. 5d). Furthermore, control birds developed less variable syllables, while conversely birds which listened to tutor song during inactivation of the LC-NCM circuit kept singing noisy syllables in adults (higher entropy value and pitch goodness, Fig. 5e), similar to that of birds isolated soon after hatching[21]. However, LC inactivation did not change the acoustic structure of innate vocal patterns, Tets, Stacks and Cackle calls, as the entropy value and pitch goodness of those calls were not different over the development between control and LC inactivated juveniles (Supplementary Fig. 5a, b), suggesting that LC inactivation did not lead to motor deficits. Together, these results indicate that LC inputs to the NCM are necessary for juveniles to learn appropriate song quality from vocal communication with a tutor, which we propose as an underlying mechanism for effective song learning via authentic social interactions.

## Discussion

Here, we demonstrate the modulation of LC-NCM neural circuit activity in juvenile zebra finches exposed to live but not artificial tutor singing. Inactivation of the LC projection to NCM during tutor singing prevented the NCM from developing song-selective neural responsiveness and juveniles from song learning. We term this mechanism "social authentication" to describe the neural encoding requirement for a live socially-interacting, e.g., "authentic", adult tutor. Functionally, we propose that an authentication circuit does not process high-resolution acoustic vocal information, like the auditory cortex, but, in contrast, monitors or "authenticates" the live social context during learning, as our analysis of LC neural activity indicates.

The noradrenergic LC is known to modulate synaptic activity in neuronal circuits involved in the learning of social communication. In mammals, the LC has been linked to brain state neuromodulation based on situational attention (e.g., fight or flight) and arousal, and LC neurons broadcast NE signals throughout the cortex in mice, rats, monkeys, humans, and zebra finches, and modulate perception of a variety of sensory stimuli[7–11,16,17,22]. Exposure to a live tutor is a multi-

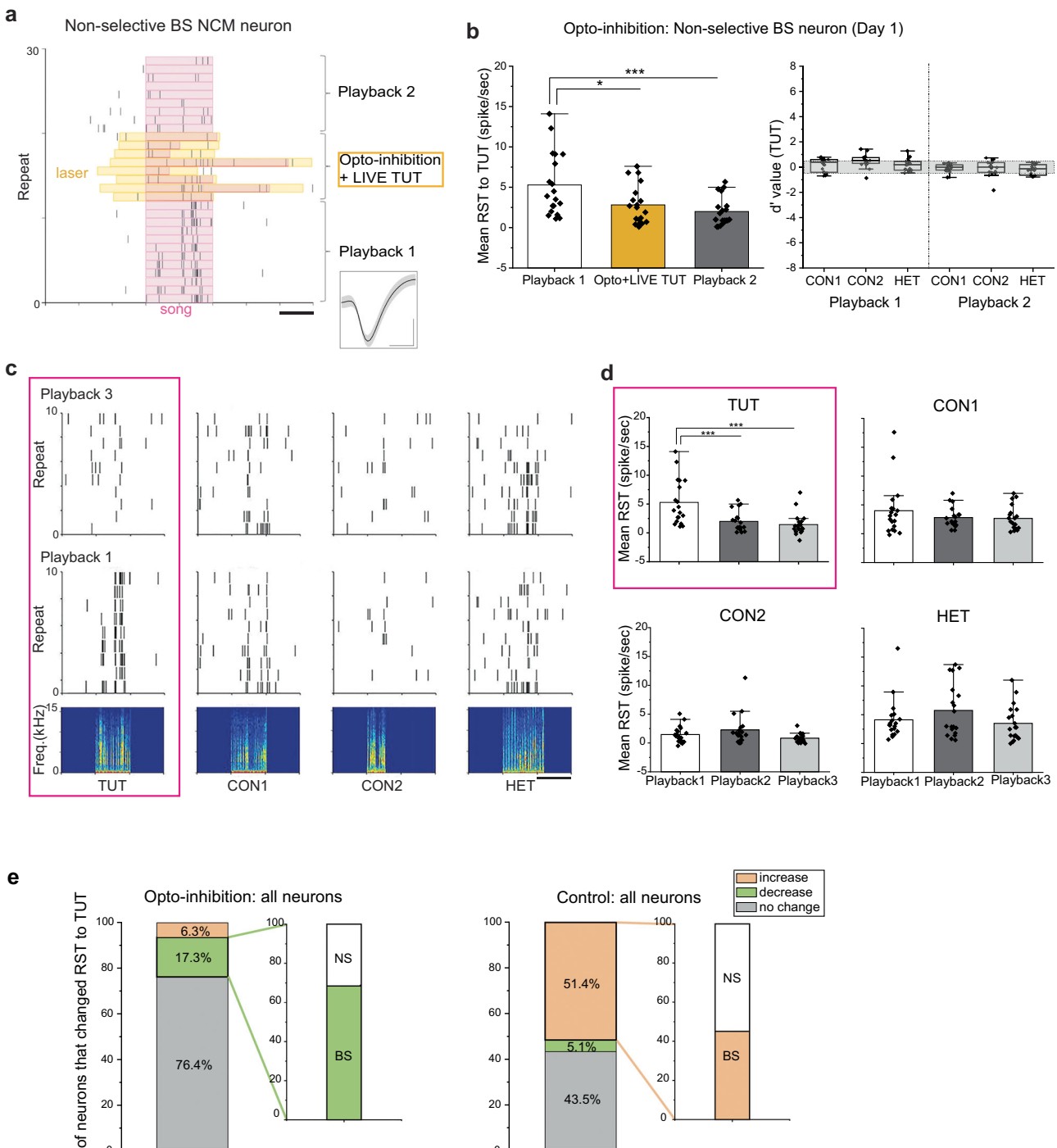

**Fig. 4 | Inhibition of LC neuronal activity during social interaction with a singing tutor diminishes auditory responses to the tutor song in a subset of BS NCM neurons. a, c** Raster plots of spiking activity in a non-selective BS NCM neuron to tutor song playback 1 and 2, or tutor singing with optogenetic inactivation of LC inputs (Opto+LIVE TUT) (**a**, scale bar: 1 s) or to playbacks of different songs before (playback 1) and after (playback 3) Opto+LIVE TUT (**c**, scale bar: 2 s, song spectrograms shown in the bottom). Inset: spike waveform of the same non-selective BS neuron (**a**, mean ± s.d., scale bars: 0.5 ms horizontal, 0.5 mV vertical). **b** Left, **d** mean response strength (RST) of non-selective BS NCM neurons to tutor song playback (Playback 1 and 2), Opto+LIVE TUT (**b**) or playback of different songs before (Playback 1) and after (Playback 3) Opto+LIVE TUT (**d**). **b** Right, Mean d-prime values for TUT over other song stimuli of non-selective BS neurons before (Playback 1) and after (Playback 2) Opto+LIVE TUT. The boxes show the 25–75%, the

center lines are defined by the median and open squares by the mean. The whiskers include all data points within 1.5 IQR (Interquartile range) and the 'outsider' dots are the data points that fall outside the whisker line. Gray areas indicate non-selective responses (−0.5 < d' value < 0.5). N = 6, n = 19 (**b**, **d**). **e** Proportion of NCM neurons that show an increase or decrease in their RST to tutor song playback after hearing LIVE TUT with optogenetic inhibition of LC (Opto-inhibition, left) or control laser stimulation (Control, right) in the NCM. N = 6 and 6 (Opto-inhibition and Control, respectively), n = 110 and 138 (Opto-inhibition and Control, respectively). BS: broad-spiking neuron, TUT: tutor song, CON1: conspecific song 1, CON2: conspecific song, HET: heterospecific song, N: number of birds, n: number of neurons. Mean ± s.e.m., *p = 0.023, ***p < 0.001, Two-sided Mann–Whitney Rank Sum Test (**b**, **d**). Source data are provided as a Source Data file.

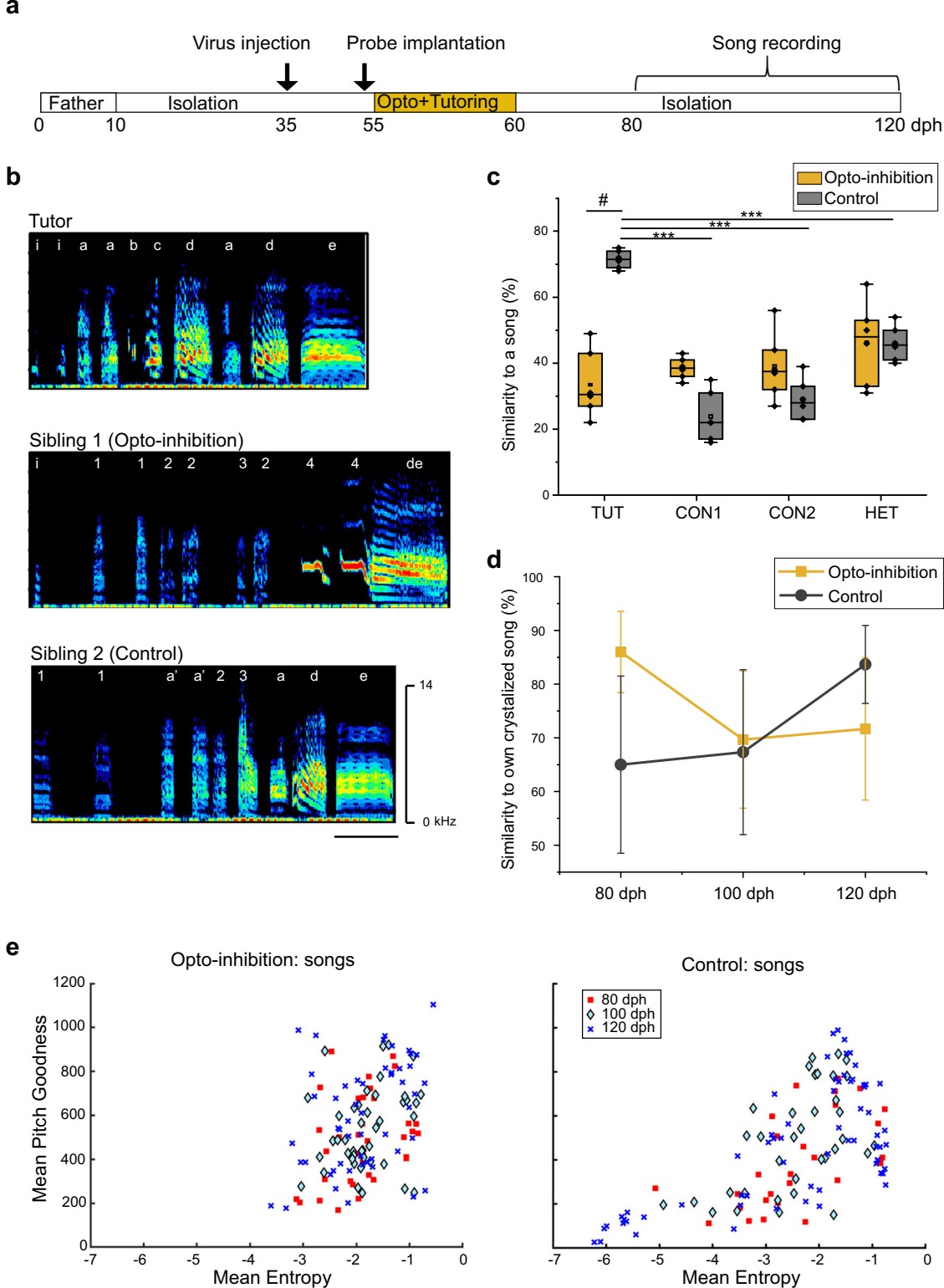

sensory experience involving auditory, visual, tactile, and olfactory modalities. Our present data raise the interesting question of how the LC integrates and authenticates multi-sensory social information for cortical broadcasting to the NCM. We found that the LC and NCM receive common anatomical inputs from an auditory thalamic region, the nucleus ovoidalis (Supplementary Fig. 6), suggesting that the NCM circuit receives feedforward, possibly neuromodulatory, inputs

through the LC when juveniles maintain social communication with a tutor. As LC neuronal responses to a tutor's song were not dependent on acoustic features of the song while on contrast the neurons in the auditory area upstream to NCM, such as Field L, are responsive to song acoustic features[23], we suggest the NCM is a candidate area that could integrate both social and acoustic information of live tutor singing for song memory.

**Fig. 5 | LC-NCM neural circuit regulates efficient song learning from social interaction with a live singing tutor. a** Experimental timeline over development. **b** Song spectrograms of tutor song (top), adult songs of two sibling birds. Sibling 1 (opto-inhibition): LC inputs were optogenetically inhibited when hearing tutor singing. Sibling 2 (Control): NCM neurons were laser stimulated when hearing tutor singing (scale bar: 0.2 s). **c** Mean song similarity score of adults to each song stimuli in birds whose LC inputs were optogenetically inhibited (Opto-inhibition) or in birds whose NCM neurons received laser stimulation (Control) during tutor singing. The boxes show the 25–75%, the center lines are defined by the median and open squares by the mean. The whiskers include all data points within 1.5 IQR (Interquartile range). **d** Song similarity score to a bird's own crystallized adult song at each point during song development after hearing the tutor song in birds whose LC inputs were optogenetically inhibited (Opto-inhibition) or in birds whose NCM neurons received laser stimulation (Control) during tutor singing. **e** Scatter plots of syllable mean entropy and mean pitch goodness throughout song development in birds with LC opto-inhibition (left) and in control birds (right), with each dot indicating a single syllable. $N = 6$ and 6 (Opto-inhibition and Control, respectively, **c**–**e**). TUT: tutor song, CON1: conspecific song 1, CON2: conspecific song, HET: heterospecific song, *N*: number of birds, dph: day post hatched, "***" indicates differences within the same group, "#" indicates differences between two groups of animals, mean ± s.e.m., #$p = 0.00000533$, ***$p = 0.0000000514, 0.0000000277, 0.00000108$, Two-sided Student T-test. Source data are provided as a Source Data file.

Our observation of long-term effects of NE modulation on NCM firing is notable. LC optogenetic inactivation during live tutor singing but not for other social interactions altered auditory responsiveness in the NCM that was selective to the authenticated tutor song for several hours. Synaptic mechanisms for NE signaling on short-term plasticity in postsynaptic cortical neurons[24] or LTP[25,26] are well known, and LC-derived NE/DA signaling in the hippocampus has been suggested to enhance memory consolidation through LTP. Here we showed that direct inhibition of LC activity during hearing of the LIVE TUT prevents song learning from the tutor, and further suggesting an LC contribution to long-term memory formation. Zebra finch NCM neurons change their firing patterns and auditory responsiveness on infusion of either NE or an α-adrenergic receptor antagonist[12,17,27,28] chiefly by decreasing their spontaneous firing rate and increasing the precision of auditory responses to songs[12,28]. We show that social communication with a tutor modulates long lasting cortical (NCM) auditory responsiveness in the first two days of social learning from a tutor, probably thorough LC inputs, but did not influence the neurons which already acquired selective auditory responsiveness. In contrast, LC neurons did not acquire TUT selective responsiveness and showed greater responses to LIVE TUT even later than the second day of tutoring. Those suggest that social tutor song experiences trigger NE release from LC neurons constantly over the development, while the released NE from LC might cause long-term plasticity to form a memory of tutor song in a specific subset of postsynaptic (NCM) neurons which might express specific NE receptors in auditory learning phase during development. Our current study does not clarify which type or pattern of neurons in the LC project to the NCM. Further studies on the anatomical connection between LC-NCM and NE release, receptor expression in various neuronal types in the NCM, and synaptic plasticity in the NCM would help to elucidate the underlying neuronal circuit mechanism of how NE neuromodulation results in acquiring selective auditory responsiveness in specific neurons.

Our findings complement and extend a recent zebra finch study demonstrating that signaling by dopamine (DA), another monoamine neuromodulator, from the midbrain PAG to a sensorimotor area, HVC, during live and not artificial song tutoring facilitates song copying[5]. Both studies show the requirement of a live tutor for vocal learning but at different time scales and anatomical stages of the ascending auditory sensorimotor pathway: DA via the PAG-HVC circuit mediates acute perception in HVC consistent with its role in vocal production, while NE in the LC-NCM circuit modulates long-lasting perception for song auditory memory. Together, the two studies suggest the evolution of a subcortical system for social authentication, with DA and NE signaling working in parallel to continuously monitor the quality of social learning for refinement of a song's acoustic features for accurate song memory and cultural transmission.

The recent literature describes a distributed cortical network for social communication in primates[29,30]. Our results in songbirds annotate this emerging framework, by exploring a specific neuronal circuit for authenticating social information in vocal learning. From an evolutionary perspective, animals such as songbirds that communicate in complex and noisy natural environments may have developed neural mechanisms to validate social interactions with species-specific individual tutors to ensure accurate song transmission for survival and reproduction. For example, the "song sharing" hypothesis suggests that female birds prefer simple, accurate songs with a well-defined lineage or geographical area[31] where tutor authentication during singing may be an adaptive mechanism. Future studies should address whether NE-dependent social authentication in the LC-NCM circuit facilitates long-term memory encoding of an individual tutor and its unique song, and whether similar mechanisms may apply in human speech acquisition

## Methods
### Animals and experimental design
Experiments were conducted following the experimental protocols approved by the Animal Care Committee at Okinawa Institute of Science and Technology (OIST) Graduate University. Thirty-four male zebra finches hatched and reared in our colony (14L: 10D light/dark condition) were used in these experiments. All birds were raised in cages with their parents and siblings until 10–12 days post-hatch (dph) when their fathers were removed. Juveniles were subsequently raised with their mothers and siblings in a cage placed in a sound-attenuating chamber until either 54–56 dph when they underwent surgery for the electrode implantation into the NCM or LC, or until 33–35 dph when viral vectors were injected into the LC. The juveniles, which were injected with viral vectors, were raised with a mother and siblings until 54–56 dph when they underwent surgery for opto-electrode implantation into the NCM. Juveniles that were implanted with an electrode or an opto-electrode, were housed individually in a sound attenuation chamber until adulthood (after 120 dph) when they were sacrificed. After recovering from the surgery (after ~24-48 h), single-unit neuronal activity was recorded during exposure to song playback stimuli for three or four consecutive days, then for the next three days a tutor was introduced into the cage, and juveniles were exposed to live tutor singing (LIVE TUT) followed by playback of the same song stimuli. Juveniles that were implanted with an opto-electrode in the NCM received laser pulse stimulation when a tutor was singing.

### Virus injection
The viral mixture of pENN-AAV-hSyn-Cre-hGH (addgene viral prep #105555-AAV9, a gift from James M. Wilson) and AAV-FLEX-Arch-GFP (addgene viral prep #22222-AAV9, a gift from Edward Boyden) or AAV-pCAG-FLEX-EGFP-WPRE (addgene viral prep #51502-AAV9, a gift from Hongkui Zeng; in a ratio 1:3) was unilaterally injected into the LC (100-180 nL) in an isolated male zebra finch juvenile (33-35 dph) through a pipette connected to a pressure injector (Nanoject II; Drummond Scientific Company, Broomall, PA, USA) with stereotaxic coordination (LC: head angle: 27°, AP: −0.3 mm, ML: 0.9 mm, Depth: 5.8–6 mm, relative to the center of Y-sinus) under isoflurane anesthesia (2.5-2.7%). After approximately 3 weeks (54–56 dph), AAV-injected juveniles were subjected to experimentation.

## Surgery and electrophysiological recordings

Single-unit neuronal activity was recorded from the LC in 12 freely behaving male juvenile zebra finches. A single Tungsten electrode (WE3PT32.0A3, MicroProbes) connected to a microdrive was implanted with stereotaxic coordination (head angle: 27°, AP: −0.3 mm, ML: 0.9 mm, DV: 5.7–5.9 mm, relative to the center of Y-sinus) and fixed to the skull with dental cement (Super-Bond C&B kit, Sun Medical, JAPAN) under isoflurane anesthesia (2.5–2.7%). Another subset of zebra finch juveniles ($n = 5$) was implanted with a 16-channel silicone probe (Buzsaki16-CM16LP, NeuroNexus) in the NCM with stereotaxic coordination (head angle: internal 45°, AP: 0.2 mm, ML: 0.5 mm, DV: 1.5–1.9 mm, relative to the center of Y-sinus) in the same manner described above. After recovery from the surgery (-24–48 hrs) the juvenile was placed in a recording arena and connected to a headstage (HST/8o25-GEN2-10P-G1-xR for LC or HST/16o25-GEN2-18P-2GP-G1 for NCM, (Plexon)) for neuronal activity recordings (56–60 dph). During song playback presentation, neuronal recordings were performed -1 h per day, 3 × 20 min for 10 repetitions of each song, for four consecutive days. During live tutor singing, neuronal recordings were performed 3–4 h per day, 20 min for 10 repetitions of each song playback followed by a 30 min break and then 1–2 h of a tutor present in the arena with occasional singing (10–20 min) followed by another 30 min break and a 20 min song playback presentation, for four consecutive days. Song playback stimuli were edited using a custom MATLAB code (MATLAB 2018b, MathWorks) and presented in a pseudo-random order using a custom LabVIEW code (LabVIEW 2016 64-bit version, National Instruments) and Data Acquisition device (NI USB-6341, National Instruments) which is used to split playback stimuli information and feed it both into an analog input channel of the OmniPlex System (Plexon) and the speaker amplifier (Topping TP21 T-Amp Class T Mini Amplifier).

Neuronal signals were amplified 10,000–20,000-fold, band-pass filtered at 0.5–9 kHz digitized (40 kHz) with the OmniPlex System (OmniPlex Software, PlexControl, Plexon). Song stimuli were played back from a loudspeaker located on the top of the arena, and all vocal activity was recorded through a microphone (lavalier microphone, C417 PP, AKG) together with the neuronal recordings.

After completion of electrophysiological recordings and subsequent song recording, electrical lesions were made (10 mA for 10 s), and recording sites were histologically confirmed.

## Optogenetical inhibition of LC neuronal activities

For NCM recording combined with optogenetics, we implanted into the NCM a 16-channel silicone probe (Buzsaki16-CM16LP, NeuroNexus). Optic fibers were attached to patch cords (FCMH2-FCL, 2×2 MM coupler 50:50 200um 0.39NA FC/PC-LC ferrule, Thorlabs) during neuronal activity recordings (56–60 dph). Neuronal recordings were done 5–6 h per day, 20 min for 10 repetitions of each song playback followed by a 30 min break and then 1–2 h of tutor presence with occasional singing combined with laser pulses (10–15 mW, 589 nm Yellow DPSS Laser with Fiber Coupled, Shanghai Laser & Optics Century Co.), followed by another 30 min break, a 20 min song playback presentation, 90 min break and a 20 min song playback presentation, for three consecutive days. For optogenetically inhibiting the neuronal activities of LC axon terminals in the NCM during live tutor singing, when a tutor started to sing with the first introductory notes (starting of a bout), a 3 s laser pulse was applied manually using Master 8 (Eight Channel Programmable Pulse Stimulator, MicroProbes) connected the DPSS Laser and a digital input of the OmniPlex System. If the tutor singing continued when a 3 s laser pulse ended, another 3 s pulse was applied, and so on. The timing of laser pulse application was recorded together with electrophysiological data. Birds behavior and vocalizations were continuously recorded and monitored using a camera and a microphone installed inside the cage and connected to Ulead Video Studio and Avisoft-RECORDER (Avisoft Bioacoustics) software, respectively.

## Song recording and analysis

Songs of experimental juveniles were recorded in a sound attenuation chamber using an Avisoft-RECORDER (Avisoft Bioacoustics) through a microphone (lavalier microphone, C417 PP, AKG) connected to an audio interface (Fast Track Ultra 8R, M-AUDIO) then to a PC. The TUT, CON1, CON2, and HET songs were also recorded and edited for use as song stimuli in the electrophysiological experiments, using Avisoft-SASLab Pro (Avisoft Bioacoustics) software. To evaluate the degree of vocal learning after tutoring, songs of experimental juveniles were recorded at 80, 100, and 120 dph (for 2–3 days), and the similarities of their songs and a tutor's songs were measured (% similarity) using Sound Analysis Pro 2011[20]. For each bird, 10 song motifs of songs at each time point were measured for similarities to the tutor's song motif and averaged for song playback during the electrophysiological recordings. Song motifs were first segmented into separate syllables based on amplitude changes and frequency. Several acoustic features were quantified: pitch, pitch mean frequency, peak frequency, and goodness; Wiener entropy and syllable and inter-syllable interval duration. Similarity to tutor song was measured between the tutor and the tutee song motifs following an automated procedure in Sound Analysis Pro 2011 that quantifies the acoustic similarity between two song motifs based on pitch, goodness of pitch, FM, AM, and Wiener entropy. Using default settings of Sound Analysis Pro 2011 such as asymmetric comparisons of mean values, minimum duration (of 10 ms) and 10 × 10 comparisons, the song similarity was calculated and the similarity percentage was used for further statistical analysis. To evaluate stability for each separated song syllable or each different call (tets, stacks or cackles) at 80, 100, and 120 dph, we measured syllable or call Mean Entropy and Mean Pitch Goodness using Sound Analysis Pro 2011.

## Retrograde tracing of LC neuronal projections

Cholera toxin subunit B (CTB) Alexa Fluor 488 or 555 conjugates (Thermo Fisher Scientific) was unilaterally injected into the NCM or LC (0.2% w/v, 50–100 nL), respectively, of three isolated male zebra finch juveniles (52–55 dph) through a pipette connected to a pressure injector (Nanoject II; Drummond Scientific Company, Broomall, PA, USA) with stereotaxic coordination under isoflurane anesthesia (2.5–2.7%). After 3–5 days birds were anesthetized and subjected to the histology protocol.

## Histology

After experimentation, birds were deeply anaesthetized with Somnopentyl and perfused with saline and then with 4% paraformaldehyde. Parasagittal brain sections were made (50 μm thickness) using a microtome (RETORATOME REM-710, Yamato). For immunostaining, slices were incubated with primary antibodies of a mouse anti-tyrosine hydroxylase antibody (1:1500, #22941, Immunostar), rabbit-anti-dopamine beta-hydroxylase antibody (1:1500, #22806, Immunostar) or rabbit-anti-GABA antibody (1:500, #A2052, Sigma-Aldrich) in PBS-T (containing 0.3% Triton-X in PBS) for 48 h at 4 °C. After washing with PBS, slices were incubated with secondary antibodies of a goat anti-mouse antibody conjugated with Alexa 568 (1:400, A11031, Thermo Fisher) or a goat anti-rabbit IgG antibody conjugated with Alexa 568 (1:400, A11036, Thermo Fisher), for 48 h at 4 °C. The slices were mounted (Fluoromount, Diagnostic BioSystem) and then subjected to imaging by confocal microscopy (LSM 780, Zeiss).

Arch-GFP/GFP positive axon terminals were quantified in the first six to eight medial sections: every odd section was immune-labeled with the rabbit-anti-dopamine beta-hydroxylase (DBH) antibody, while every even section was immune-labeled with the combination of mouse anti-tyrosine hydroxylase antibody (TH) and rabbit-anti-GABA antibody. Microscope pictures at 40× magnification of four fields within the NCM area were taken from each section and used to quantify the percentage of DBH, TH, or GABA-double labeled axon

terminals within the Arch-GFP/GFP-positive ones, using Fiji (ImageJ) software package. Those numbers were averaged within a section and across birds within a comparing group, six birds in Opto-inhibition and six in the Control group.

### Electrophysiological data analysis

Spike sorting was performed off-line using the Offline Sorter v3 (Plexon), and well-isolated single units were submitted to subsequent analysis with NeuroExplorer v5 software packages and custom MATLAB codes (MATLAB 2018b, MathWorks). LC neurons were classified into regular and fast spiking neurons based on their waveform shape and firing rate. For each unit, we calculated the full width at half maximum of the valley portions of the average spike and the spike duration defined by the time from peak to valley[21]. We calculated the spontaneous firing rate by averaging the number of spikes during a 50 min period when no sensory stimulus was presented. LC neurons with spontaneous firing rates < 20 Hz were classified as regular-spiking while neurons > 20 Hz were counted as fast-spiking. NCM neurons were classified into broad or narrow spiking neurons based on the mean spike width and the duration from negative to positive peak[16]. For both NCM and LC neurons we quantified auditory responses of each neuron by response strength (RST): the difference in mean firing rate during the song stimulus ($FR_{stim}$) and the firing rate during the same duration period with the song stimulus just before the stimulus ($FR_{base}$) with the following formula:

$$\text{Response strength (RST)} = FR_{stim} - FR_{base}$$

To measure the response bias between two song stimuli, we calculated d-prime values using the equation:

$$D - \text{prime}_{a-b} = \frac{2(\overline{RST}a - \overline{RST}b)}{\sqrt{\sigma_a^2 + \sigma_b^2}}$$

where $\overline{RST}$ is the mean response strength to the stimulus and $\sigma^2$ is the variance of the RST[25]. D-prime value > 0.5 was used as a criterion for biased response. If d-prime value comparisons between the TUT song and all other songs were greater than 0.5, the neuron was categorized as selective for the TUT song.

### Statistical analysis

All statistical analyses were conducted using the SigmaPlot 13.0 software package. After conveying Normality and Equal Variance tests for all compared groups, we conducted either a Student's T-test or a Mann–Whitney Rank Sum Test. Additionally, for data consisting of Playback 1, 2, and 3 and different auditory stimuli such as TUT, CON1, CON2, and HET groups, the two-way ANOVA followed by Holm–Sidak post hoc test was performed, using 'Playback' as the first and 'Auditory stimulus' as the second factor with All Pairwise comparisons. All comparisons were considered significantly different if $p < 0.05$. For data visualization Origin 2019b software was used.

### Reporting summary

Further information on research design is available in the Nature Research Reporting Summary linked to this article.

## Data availability

Source data are provided with this paper. The data that support the findings of this study are included in the Supplementary Table 1. The datasets generated during this study and any additional information required to reanalyze the data reported in this paper are available from the corresponding author upon reasonable request. Source data are provided with this paper.

## Code availability

Custom codes employed for the data collection and analyses during the current study are deposited and available at: https://doi.org/10.5281/zenodo.6630127, https://doi.org/10.5281/zenodo.6630093 and https://doi.org/10.5281/zenodo.6630340.

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

## Acknowledgements
We thank Dr. C. Yokoyama for valuable help in preparing the manuscript, and Drs. R. Mooney and S.C. Woolley for critical reading of the manuscript. We appreciate Dr. M. Araki for MatLab coding assistance and Mrs. A. Kuneji for taking care of the experimental animals. We also thank Mr. Nicolas Baudoin for providing bird images for figures. This research was supported by OIST Graduate University and the JSPS KAKENHI JP grants: #19K16302 to J.K. and #18H02531 and #20H05075 to Y.Y.-S.

## Author contributions
Y.Y.-S., J.K., and Y.M. designed the experiments; J.K. performed the experiments and analyzed the data; Y.Y.-S. and J.K. wrote the paper.

## Competing interests
The authors declare no competing interests.
