## [Peer Review File · Nature Communications]

Neural Circuit for Social Authentication in Song LearningREVIEWER COMMENTS

Reviewer #1 (Remarks to the Author):

The authors employ multi-unit recordings and optogenetics manipulations to describe how the locus coeruleus (LC) increases activity and selectivity in the neural circuit of song learning, particularly in the caudomedial nidopallium area (NCM), with direct behavioral consequences in the adult zebra finch. The authors found that LC increases its activity during social interaction with a tutor, which persists during the second playback of the tutor song. A similar pattern is observed (as was reported before) for NCM neurons. As the authors have shown previously, NCM neurons not only increase their activity but also develop specificity for the tutor song after exposure to live tutoring. The authors then argue that LC must play a role in shaping this form of plasticity, and perform optogenetics inhibition of LC axons locally in NCM to verify this hypothesis. They found that inhibiting LC axons during tutoring decreases the number of neurons that develop song specificity, and these manipulations have negative long-term effects in adults.

The paper is well written and follows a clear narrative, but suffers from confusing figure organization and interpretation of the data. While the observations are exciting, there is a major lack of mechanistic explanations which could be tested experimentally (or at least conceptually in the discussion). Is the LC truly the modulation center that serves to sharpen responses in NCM, or a relay center which could cause the observed effects when inactivated? It is also unknown how the LC acts on NCM neurons, and why does it target the specific subpopulation of BS neurons? Are those neurons known to express specific NE-receptors, or is it mediated by another circuit element such as astrocyte or local interneurons? The authors also do not convincingly demonstrate that LC inputs to NCM are noradrenergic. The rapid spiking onset to laser stimulation would rather argue for a GABAergic disinhibitory effect. These issues need to be carefully addressed by the authors before further recommendation can be made.

Here are my more detailed comments for each part of the manuscript:

Figure 1:

Major:

1. Panel F should be moved to A as it describes the behavioral paradigm used throughout the manuscript. The corresponding text in the manuscript should be reorganized.
2. Panel A-C: I am not sure why the authors have chosen to display this information here. It is not crucial for understanding the paper or the rest of the figure, and it could be moved to Extended Data instead.

3. Panel G: It says there are 16 neurons from 8 animals. But in the text it's written that 29 neurons were recorded from 12 animals and in C, there seems to be 14 RS and 15 FS neurons. Which one is correct? Also please show all the data points for all graphs (also for Extended Data 2C, etc).

4. It is not shown whether LC neurons have or develop specificity to the tutor song after live tutoring like NCM (figure 2-3) does? It is intriguing that LC neurons elevate their activity without specificity. Also, is there any difference between FS and RS neurons?

5. ED-4: In one animal exposed to a tutor which did not sing, the authors do not see an increase from playback 1 to playback 2. But do they see an increase in LC activity during the social interaction?

6. Panel I-J : Those are perhaps the most mysterious data in the manuscript. There is no quantification of the fluorescence or level of arborization of LC axons into NCM, no number of replicates given, no control where DOX is given for 3 days to animals which did not undergo social interaction. If the point of these panels is to establish that LC projects to NCM, why not alternatively just quantify the amount of DBH-positive vs EYFP-positive axons in NCM, as well as the overlap of the two, in birds injected with simply EYFP into the LC?

Minor:

7. Panel B-C: clustering of FS and RS neurons is visibly not ideal. Those criteria may not be optimal to separate the two groups. Have the authors tried to use different criteria (burst index, slope, ratio of peak to trough, etc, ...) or approach (PCA)? Why is 20Hz the threshold for FS neurons?

8. Line 76: "there was substantial variation in the pattern of LC neuron activity". According to Extended Data 1, LC neurons seem to fire consistently high across all the repeats. I'm not sure what the "substantial variation" refers to here?

9. ED 1c: It is interesting that this FS LC neuron stops firing upon introduction of a tutor in the cage. Is it the case for all FS neurons? Can this be quantified? How long does the decrease last? Does it correspond to an overshoot in RS neurons? Would this mean that there is direct disinhibition to allow RS LC neurons to increase their activity during social interaction?

Figure 2:

Major:

10. Panel B: is this axon also DBH-positive? It is quite important to know which population of LC neurons project to NCM: FS (putative GABAergic) or RS? Optogenetic inactivation seems to immediately evoke spiking in the example NCM neurons shown in C, which would argue for a direct release of inhibition rather than noradrenergic modulation. The authors need to do a better job at quantifying this information.

11. Panel F: does this figure reflect the fact that a lot of NCM neurons do not develop specificity to the tutor song in playback 2? Or is the reduction in RS also present for the small fraction of NCM neurons which either keep or develop specificity?

12. Please show all the single data points.

Minor:

13. Panel F: I can't differentiate the greyed vs non-greyed bars. Could you use solid vs empty bars instead?

14. Line 133: "suggesting that activation of LC inputs to the NCM by a live, singing tutor enabled NCM neuronal circuits to acquire song-selective auditory responses." This is not quite accurate as the authors perform inactivation which decreases song-selectivity, but the reverse is not necessarily true.

Figure 3:

Minor:

15. Panel D: Can the authors also show the fraction of song-selective NS neurons? Although they have previously shown that they don't develop selectivity after live tutoring, it would be good to show that these results are replicated here.

Figure 4:

Major:

16. ED 5 is very interesting and should be moved to the main figure.

Minor:

17. Line 175: "caused by a loss of motor plasticity". What does this mean, as the authors so far have only investigated auditory processing?

18. Panel C: How is the pound sign (#) different from the asterisks? It seems that both refer to <0.005 according to the legends?

Others major comments:

19. Extended Data 6: No quantification or replicates (see comment #6)

20. As stated above, please show all single data points when bar graphs are used.

Reviewer #2 (Remarks to the Author):

In this study (Katic et al, Neural Circuit for Social Authentication in Song Learning), authors investigate the role of functional neural connections between Locus coeruleus (LC) and Caudomedial Nidopallium (NCM) in the social authentication of song-learning in juvenile Zebra finches. Building on previous knowledge in this field, authors take an important step forward to improve our understanding of the neural basis of necessity for social interaction during effective song-learning. I am especially excited by single-cell recordings and the optogenetics experiments as they provide crucial evidence to narrow down the role of LC-NCM neural circuits in social song-learning. The findings presented in the study speak to a broad audience while being a timely extension of knowledge in the field of social learning. Considering the study in the context of today's world when virtual learning is becoming mainstream, I believe such results can motivate further quantitative investigations of differences across learning in virtual and real-world environments. I enjoyed reading the manuscript as it is engaging, clearly written, and the information flows well from the beginning to the end, however, I also think there are some aspects in the study that need some revision to improve the clarity and make the results more accessible to the readers.

Major concerns:

- Comparison of mean response strength to TUT playback1 for BS-NCM neurons in Fig. 1h and Fig. 2h: In Fig.1h, twelve BS-NCM neurons show mean RS close to 0 spikes/sec for TUT-playback 1, whereas, in Fig. 2h, nineteen BS-NCM neurons show mean RS of 5 spikes/sec for TUT playback 1. The authors should discuss the possible causes of variability in mean RS of BS-NCM neurons for TUT playback1 before Live-tutor interaction in these two groups when all other conditions appear to be the same.

- Authors should include the raw data points (response strengths for individual neurons) for all the bar plots showing mean response strength. For example, Fig 1g, 2d, 2f,2h, Extended data 1f, Extended data 2C, Extended data 3a,b - Raw data points (number of neurons) should be shown overlapping with the bar plots.

- Authors should include a summary table with information about firing rates for all the neurons during playbacks and live-tutor song experience to address some visualization inconsistencies such as between Line 74-75 and Extended Data Fig. 1a - For some of the trials, fast-spiking neurons seem to decrease their response to song playback as compared to their spiking before the song playback, such as for CON1 - Trial # 4-6, HET- Trial # 5, CON2 - Trial#4.

- Line 120-123 - In this section, authors should also mention the actual number of neurons in addition to the percentage of neurons.

- Readers who might be working with other animal models such as rodents/primates may consider it an issue that there is a seemingly low number of neurons in the data for some figures to draw the conclusions. Early in the main text, authors should mention the total number of neurons recorded along with the fractions of silent and song-selective neurons for each figure. Authors may also acquaint the readers with the practical difficulties in performing single-unit recordings from LC and NCM in zebra finches. Authors may clarify to the readers that such numbers of neurons as recorded in this study are usual with zebra finches, if that is the case. For example, authors may provide a table, summarizing data from all the previous studies that have recorded single neurons with zebra finches. In the table, authors can include the brain regions (LC, NCM, and other brain regions) in one column, corresponding numbers of single units recorded from zebra finches in the second column, and relevant references in the third column.

- Line 181-182 and Extended Data Fig. 5b - For the left panel, the range of mean entropy is narrower for the opto-inhibited group, indicating lesser variability in the syllables produced by the opto-inhibited group as compared to the controls. Even though the range of syllables produced by the opto-inhibited group is similar to the birds isolated soon after hatching, it does not exclude the possibility of a loss of motor plasticity from LC inactivation. Authors should provide additional references to justify that LC inhibition does not lead to major motor deficits.

- Extended Data Fig 4. Not sufficient data to make the claims associated with fig 4. Authors may mention these findings in the discussion as the statistical power of these data seems insufficient to present them along with main results, unless they have some additional data to make these points.

- Line 557 - Authors should clarify how RS was estimated, preferably with a formula. The following is not clear to the reader: what were the durations of sound stimulus and pre-stimulus baseline period which were used to calculate the mean firing rate for estimating response strength.

- Line 522 - Reference 39 seems to be missing from the list. Authors should expand SAP2011 and provide a brief summary of how the song similarity was calculated using SAP 2011.

Minor concerns:

- Figure 1c. Red bars appear overlapping with non-uniform widths. Bar-width or x-axis can be readjusted to improve clarity in the figure

- Fig 1d - It is an intriguing observation that TUT selective LC neurons stop their firing for 2.5 seconds when a tutor is introduced in the cage. The authors may further discuss the possible causes and effects

of the cessation of firing in TUT-selective LC neurons. Do all the TUT-selective LC neurons stop firing reliably for 2.5 sec when a tutor is introduced? It would be interesting to check how the activity of NCM neurons varies around tutor introduction in opto-inhibition experiments (Fig 2) as compared to their activity in Fig 1.

- Figure 1f. 'dph' should be expanded in the figure legend

- Line 67: The sentence should include the information about the 30 min gaps between Playback 1-Live-tutor-playback 2. (For example, "...minutes of passive playback (Playback 2) at 30 min intervals each".)

- Line 67: Recent evidence in rodents - 's' is missing from rodents.

- Line 121 - The sentence could be restructured to improve readability (For example, "However, a subset of otherwise silent NS neurons was activated only when...")

- Methods: Information about the light-dark cycle for housing the subjects should be added.

- Line 449 - Authors should add a reference for the dox dosage.

- Line 476 - Stereotaxic coordinates for virus injection should be added to the methods.

- Line 488 - How many birds were implanted with a 16-ch silicon probe?

- Line 494-495 - Is this data presented in the results?

- Line 504 - What was the approximate intensity of the 3s laser pulse?

- Fig 3a - bar plots should be added

Aditya Singh

Reviewer #3 (Remarks to the Author):

In this manuscript, the authors use optogenetic methods show that projections for the Locus Coeruleus (LC) to the song nucleus NCM plays an important role in the social aspects of song learning. Neurons in NCM and LC respond more vigorously to song when presented by a live tutor vs. passive playback, and the response to passive tutor song showed a long-lasting increase after the tutoring sessions. However, when the projections from LC to NCM are optogenetically inhibited during the period of exposure to a live tutor, the increase in NCM activity appears to reverse. Furthermore, birds subject to optogenetic suppression during live tutoring make poorer copies of their tutor song than controls.

These results hold great promise for having a long-term positive impact on our attempts to understand the function of the LC. A key question for the field is the behavioral specificity of different LC afferents and efferents. The experimental control afforded by the songbird learning paradigm - both sensorimotor and social - may provide a powerful paradigm for investigating these issues.

However, the paper suffers from interlocking issues related to the complexity of the results and the clarity with which they are presented. Furthermore, there may be a question about whether the songs used for stimuli (both tutoring and control) are appropriately counterbalanced.

It was difficult to track all the separate experiments. In figure 2h. Response strength (RS=response-background) to Tutor, conspecific 1 and conspecific 2 and heterospecific song are shown before and after exposure to the live tutor with opto suppression of LC inputs. The conclusion from the figure is that RS goes down selectively for the tutor. But during playback 1 TUT, CON1 and CON2 are just three different zebra finch songs the juvenile has never heard before. But response strength varies quite a bit between songs, especially between TUT and CON2. If the initial response was like CON2 would you be able to obtain this result? This is a common problem in the song field in that all randomly selected songs have differences that are consistent across birds and neurons. Dealing with these issues relies on careful experimental design and counterbalancing. The authors may have enough data to rule out this as a possibility, but readers need to be able to track this down to be confident of the results.

A second major issue is the complexity of the results. The authors show intriguing differences in responses between different neuron classes defined by spike width and spontaneous activity. However, these differences are not laid out clearly. The section between line 79 and 101 is particularly dense and

describes an entire series of results with little commentary. But not a single one of these response type differences was important enough to mention in the discussion. The authors should give serious consideration to paring the paper down to the main results and discussing response-type differences in a more extensive paper. If they do want to retain cell-type differences, it may be useful to present the main results so that the reader can follow them through, and then show how these properties are distributed across different cell-types.

There are many other minor issues mostly having to do with clarity. I will cover a representative sample.

The authors state that they “characterize a neural circuit for social authentication” (14). The results only investigate the projection from LC to NCM - at best one component of the social authentication circuit. The authors show a hint of a projection from the thalamus to LC, but this is in supplemental data. Is LC the place where “social authentication is computed”, or does LC broadcast the authentication signal to modulate plasticity?

The authors distinguish their results from effects on song prosody. Issues related to timing and sequencing have been tied to song nucleus HVC and previous results show that social interaction modulates dopamine-dependent mechanisms related to HVC. Readers need to be given better explanation to understand distinctions between different aspects of song learning that are only clear to experts.

There were many phrases about “reconfiguring responses”, “disordered responses,” “modulation of activity” and effects on song learning. Examples include

- 18. “LC firing reconfigured long-term song-selective neural responsiveness”
- 20. “optogenetic inhibition of LC presynaptic signaling in the NCM disordered NCM neuronal responsiveness to live tutor singing”
- These are two section headings. Do they mean something different?
 - o 103. LC inputs configure auditory responses of NCM neurons
 - o 136 Live tutor singing modulates NCM neuron activity via LC inputs
- 157. “reorganize and maintain neuronal circuit activity in the NCM via functional inputs from the LC.”

While the authors have shown that optogenetically inhibiting LC-NCM projections disrupts learning, it is not clear whether LC is affecting activity that then alters learning, or modulates plasticity/gene expressions that effects how or how strongly NCM activity patterns are written into long-term changes, or something in between or something else. The authors should be careful about statements like LC inputs reconfigure LCM activity since this can mean many things.

51. “We observed enhanced neural activity in the LC-NCM neuronal circuit during vocal communication with a live adult tutor. Optogenetic inhibition of the LC-NCM neural circuit during exposure to a live

singing tutor reduced song learning.” The authors have shown some optogenetic reduction in activity, but it is not clear that the reduction in activity during tutor song singing is the main cause of the reduced song learning.

194. “The noradrenergic LC provides an unexpected synaptic perspective on the learning of social communication.” 208. “Synaptic mechanisms for NE signaling on short-term plasticity in postsynaptic cortical neuron (23) or LTP (24, 25) are known, but synaptic changes correlated to long-term physiological and behavioral outcomes, as we show in NCM, are rare.” I’m not in this field, but are the authors’ results really unexpected? Is it rare to show LC having long term effects on behavior? If so, please discuss further.

222. “DA and NE signaling working in parallel to continuously monitor the quality of social learning.” DA and NE signaling do not monitor the quality of social learning. There are several places in the paper where the authors could take greater care in their wording.

225. “Our results in songbirds annotate this emerging framework, adding the authentication of social interactions to brain-wide networks.” Not sure what this means.

The authors mention that juveniles can learn if they actively trigger playback. This would be a great experiment in the current context!

Figures:

It seems that the TUT song RS for playback 1 is generally quite low (<1) in 1h, 2d and 2f, but then TUT song RS is about 5 in 2h. Also, from, 2d and 2f it seems that RS stays the same or increases from playback 1 to 2 with opto, but in 2h it decreases. What is going on here?

I don’t understand 2f. I think the legend is wrong. But if I’m reading it right, the total response may stay about the same or go down, but RS goes up since baseline firing goes down? This needs explanation.

It seems that all of figure 3 and figure 2c-f are making the distinction between NS and RS NCM neurons. Seems like a lot of space given that little is made of the distinction. In 2c-f I would suggest NS and RS be somewhere in the subplot title or y axis label so that a reader doesn’t have to dig through the legend to see what distinctions are being made (like in fig 3). It’s hard to trace all the bars in 3a. Perhaps a scatter plot with playback 1 along the x axis and playback 2 along the y axis plus a dashed diagonal? That might even make it easier to convey the information in 3b (perhaps eliminate it)? It seems that 3c is kind of

redundant with fig 2 and 3d is similar to 3b. What is the bottom line here? Lot's of graphs, hard to figure out what to make of them.

Figures 1 and 4 are well done.

REVIEWER COMMENTS

Reviewer #1 (Remarks to the Author):

The authors employ multi-unit recordings and optogenetics manipulations to describe how the locus coeruleus (LC) increases activity and selectivity in the neural circuit of song learning, particularly in the caudomedial nidopallium area (NCM), with direct behavioral consequences in the adult zebra finch. The authors found that LC increases its activity during social interaction with a tutor, which persists during the second playback of the tutor song. A similar pattern is observed (as was reported before) for NCM neurons. As the authors have shown previously, NCM neurons not only increase their activity but also develop specificity for the tutor song after exposure to live tutoring. The authors then argue that LC must play a role in shaping this form of plasticity, and perform optogenetics inhibition of LC axons locally in NCM to verify this hypothesis. They found that inhibiting LC axons during tutoring decreases the number of neurons that develop song specificity, and these manipulations have negative long-term effects in adults. The paper is well written and follows a clear narrative, but suffers from confusing figure organization and interpretation of the data. While the observations are exciting, there is a major lack of mechanistic explanations which could be tested experimentally (or at least conceptually in the discussion). Is the LC truly the modulation center that serves to sharpen responses in NCM, or a relay center which could cause the observed effects when inactivated? It is also unknown how the LC acts on NCM neurons, and why does it target the specific subpopulation of BS neurons? Are those neurons known to express specific NE-receptors, or is it mediated by another circuit element such as astrocyte or local interneurons? The authors also do not convincingly demonstrate that LC inputs to NCM are noradrenergic. The rapid spiking onset to laser stimulation would rather argue for a GABAergic disinhibitory effect. These issues need to be carefully addressed by the authors before further recommendation can be made.

Thank you for your comment valuable suggestions. We are glad to know that the reviewer found our paper is exciting and well-written. For the mechanistic explanation of how the LC sharpens responses in the NCM, we have added more data showing the LC inputs were DBH/TH-positive and augmented the text discussion on the possible neuronal circuits in the NCM and the LC regulation with LIVE TUT singing. We provide responses to your individual comments below.

Here are my more detailed comments for each part of the manuscript:

Figure 1:

Major:

1. Panel F should be moved to A as it describes the behavioral paradigm used throughout the manuscript. The corresponding text in the manuscript should be reorganized.

Thank you for the helpful suggestion. We have reorganized the figures according to this suggestion along with those of the other reviewers.

2. Panel A-C: I am not sure why the authors have chosen to display this information here. It is not crucial for understanding the paper or the rest of the figure, and it could be moved to Extended Data instead.

We moved those panels to Extended Data Figure 1.

3. Panel G: It says there are 16 neurons from 8 animals. But in the text it's written that 29 neurons were recorded from 12 animals and in C, there seems to be 14 RS and 15 FS neurons. Which one is correct? Also please show all the data points for all graphs (also for Extended Data 2C, etc).

We conducted the experiments in juvenile birds and made neuronal recordings from the LC, a deep area in the brain before testing auditory responses. However, we lost 6 neurons from 3 birds after characterizing LC neuronal activity before the auditory testing. Also, for one bird (3 neurons recorded), a tutor did not sing at all for the tutoring period, the data of which were included in the previous version, but are now excluded as the number of neurons were small. So, we have 29 neurons for the Extended Fig1 a-c, while 16 neurons for Fig1 and Extended Fig1 e-g. These are now clearly stated in the text and figure legends. Also, we added all the data points in the figures.

4. It is not shown whether LC neurons have or develop specificity to the tutor song after live tutoring like NCM (figure 2-3) does? It is intriguing that LC neurons elevate their activity without specificity. Also, is there any difference between FS and RS neurons?

Thank you for the suggestion. Both the FS and RS LC neurons did not develop selectivity after tutoring as they increased RS to all song stimuli, in contrast to NCM neurons. These data are now shown in Fig 1 d-f. For auditory responsiveness we found no differences between FS and RS, but only FS showed a pause in firing upon introduction of a tutor. These points are now addressed in the text and in Extended Fig. 1d.

5. ED-4: In one animal exposed to a tutor which did not sing, the authors do not see an increase from playback 1 to playback 2. But do they see an increase in LC activity during the social interaction?

One FS neuron responded with a pause when the bird saw the tutor. Apart from that, it was difficult to track other social cues such as if the bird was looking at the tutor. Since this was insufficient data from only one bird, we decided to remove it from the current manuscript.

6. Panel I-J : Those are perhaps the most mysterious data in the manuscript. There is no quantification of the fluorescence or level of arborization of LC axons into NCM, no number of replicates given, no control where DOX is given for 3 days to animals which did not undergo social interaction. If the point of these panels is to establish that LC projects to NCM, why not alternatively just quantify the amount of DBH-positive vs EYFP-positive axons in NCM, as well as the overlap of the two, in birds injected with simply EYFP into the LC?

As we would like to confirm the projection from LC to NCM in juveniles here, we replaced it with the data showing the LC-NCM projection using viral axonal labelling. We quantified the amount of co-labelling with GABA, DBH or TH immunoreactivity and the results are now shown in Extended Data Fig. 3a.

Minor:

7. Panel B-C: clustering of FS and RS neurons is visibly not ideal. Those criteria may not be optimal to separate the two groups. Have the authors tried to use different criteria (burst index, slope, ratio of peak to trough, etc, ...) or approach (PCA)? Why is 20Hz the threshold for FS neurons?

After we performed PCA and checked other criteria, we found that the current criteria of firing rate gave a better separation. We applied the threshold of 20Hz based on: 1) previous research

in mice in which the firing rate of FS is slightly lower than in birds (16); 2) all analyzed FS neurons (higher than 20Hz firing rate) showed a pause upon introduction of a tutor, whereas RS neurons (lower than 20Hz firing rate) didn't (Extended Data Fig. 1d).

8. Line 76: "there was substantial variation in the pattern of LC neuron activity". According to Extended Data 1, LC neurons seem to fire consistently high across all the repeats. I'm not sure what the "substantial variation" refers to here?

We now rephrase this as "the pattern of LC neuron activity varied across repeated presentations of the same song" (192).

9. ED 1c: It is interesting that this FS LC neuron stops firing upon introduction of a tutor in the cage. Is it the case for all FS neurons? Can this be quantified? How long does the decrease last? Does it correspond to an overshoot in RS neurons? Would this mean that there is direct disinhibition to allow RS LC neurons to increase their activity during social interaction?

We quantified the duration of the pause and compared it across FS and RS LC neurons. We found that FS neurons showed significantly longer pauses in their firing upon tutor introduction, which is now showed in Extended Data Fig. 1d. We did not see an overshoot in the RS neurons and cannot tell if the pause of FS disinhibits RS neuron firing as we have yet to analyze the local circuits in the LC. We discuss potential LC-NCM neuronal circuit functions in the discussion.

Figure 2:

Major:

10. Panel B: is this axon also DBH-positive? It is quite important to know which population of LC neurons project to NCM: FS (putative GABAergic) or RS? Optogenetic inactivation seems to immediately evoke spiking in the example NCM neurons shown in C, which would argue for a direct release of inhibition rather than noradrenergic modulation. The authors need to do a better job at quantifying this information.

We quantified Arch-positive LC-NCM axons for their immunoreactivities to DBH, TH and GABA. We found that LC-NCM axons were mostly immuno-positive for DBH or TH, but rarely for GABA. These data are now shown in Fig 3b.

11. Panel F: does this figure reflect the fact that a lot of NCM neurons do not develop specificity to the tutor song in playback 2?. Or is the reduction in RS also present for the small fraction of NCM neurons which either keep or develop specificity?

We are sorry for the confusion. We found the proportion of tutor selective neurons did not increase when juveniles heard a tutor's song with opto-inhibition of LC-NCM which is now shown in Fig. 3g. We also found seven NCM neurons which showed selectivity to TUT in Playback 1 in the first day of tutoring. All seven neurons did not show greater responses to LIVE TUT with opto-genetic inhibition of LC inputs, while six of the seven neurons sustained selective response to TUT after being exposed to LIVE TUT (Playback2). In contrast the neurons that showed non-selective responses to TUT in Playback 1, did not respond to LIVE TUT with LC opto-inhibition and had reduced responses to TUT playback after that. These data are now shown in Fig 3 d-f and Fig 4 and described in the text (1151-173).

12. Please show all the single data points.

All individual data points are now included in the figures.

Minor:

13. Panel F: I can't differentiate the greyed vs non-greyed bars. Could you use solid vs empty bars instead?
We modified that figure and now use a different color code which we believe is more visible.

14. Line 133: "suggesting that activation of LC inputs to the NCM by a live, singing tutor enabled NCM neuronal circuits to acquire song-selective auditory responses." This is not quite accurate as the authors perform inactivation which decreases song-selectivity, but the reverse is not necessarily true.

Now we rephrase this sentence to read "Those results suggest that activation of LC inputs to the NCM by a live, singing tutor enabled NCM neuronal circuits to acquire TUT-selective auditory responses in specific neurons." (1172).

Figure 3:

Minor:

15. Panel D: Can the authors also show the fraction of song-selective NS neurons? Although they have previously shown that they don't develop selectivity after live tutoring, it would be good to show that these results are replicated here.

We found one of the NS neurons developed selectivity responses to any song in both the normal and opto-inhibition conditions. These data are now included in the Extended Data Fig 2e-g, and Extended Data Fig 3c.

Figure 4:

Major:

16. ED 5 is very interesting and should be moved to the main figure.

These panels are now incorporated in Fig 5.

Minor:

17. Line 175: "caused by a loss of motor plasticity". What does this mean, as the authors so far have only investigated auditory processing

We thought that one possible reason for not learning from tutor singing with opto-inhibition was a loss of ability to change vocalization due to motor deficits. Now the sentence is rephrased to read "To investigate if lesser similarities to tutor songs were caused by losing the ability to change vocal motor patterns with LC inactivation" (1202).

Regarding this issue, we further tested if opto-inhibition affected other vocalizations (calls) and found no effects on call acoustic features. These data are now shown in Extended Data Fig 6 and stated in the text (1209-).

18. Panel C: How is the pound sign (#) different from the asterisks? It seems that both refer to <0.005 according to the legends?

The pound sign (#) refers to the statistical differences between two different groups (opto-inhibition and control), while asterisks are for differences within a group (ex. TUT control vs CON1 control). These distinctions are now clearly stated in the figure legends.

Others major comments:

19. Extended Data 6: No quantification or replicates (see comment #6)

We examined the retrograde labelling from both the LC and NCM in three adult birds, and found the projection from the same neurons in the Ovoidalis in all three birds. These data are shown in the Extended Data Fig 7 and described in the text (1233-).

20. As stated above, please show all single data points when bar graphs are used.

Thank you for the suggestion. We now include all the data points in the figures.

Reviewer #2 (Remarks to the Author):

In this study (Katic et al, *Neural Circuit for Social Authentication in Song Learning*), authors investigate the role of functional neural connections between Locus coeruleus (LC) and Caudomedial Nidopallium (NCM) in the social authentication of song-learning in juvenile Zebra finches. Building on previous knowledge in this field, authors take an important step forward to improve our understanding of the neural basis of necessity for social interaction during effective song-learning. I am especially excited by single-cell recordings and the optogenetics experiments as they provide crucial evidence to narrow down the role of LC-NCM neural circuits in social song-learning. The findings presented in the study speak to a broad audience while being a timely extension of knowledge in the field of social learning. Considering the study in the context of today's world when virtual learning is becoming mainstream, I believe such results can motivate further quantitative investigations of differences across learning in virtual and real-world environments. I enjoyed reading the manuscript as it is engaging, clearly written, and the information flows well from the beginning to the end, however, I also think there are some aspects in the study that need some revision to improve the clarity and make the results more accessible to the readers.

Thank you for the very supportive comments. We are happy to learn that the reviewer found our study expands our knowledge on social learning and will have an impact on readers. We made revisions in our manuscript to improve its' readability. Hopefully, the reviewer will find that our data is now clearly presented. We also provide our responses to each individual comment below.

Major concerns:

- Comparison of mean response strength to TUT playback1 for BS-NCM neurons in Fig. 1h and Fig. 2h: In Fig. 1h, twelve BS-NCM neurons show mean RS close to 0 spikes/sec for TUT-playback 1, whereas, in Fig. 2h, nineteen BS-NCM neurons show mean RS of 5 spikes/sec for TUT playback 1. The authors should discuss the possible causes of variability in mean RS of BS-NCM neurons for TUT playback1 before Live-tutor interaction in these two groups when all other conditions appear to be the same.

We are sorry that the data presentation was confusing. We used the data from different neuronal types for the previous Fig1h and Fig2h. Fig1h represented the BS neurons, which acquired TUT selective responses "after" hearing LIVE TUT singing (Playback 2), while Fig. 2h was for those neurons which showed TUT selective responses "before" hearing LIVE TUT (Playback 1). That is why the average RS in Playback 1 (before LIVE TUT) was higher in Fig. 2h.

We now show the average RS to TUT in all BS neurons with all the data points from each neuron in the normal (Fig. 2a) and Opto- (Fig. 3c) condition, which showed no difference in their distribution. In addition, we again provided the data from the BS neurons which acquired TUT selective responses "after" hearing LIVE TUT singing (Playback 2) in the normal condition (Fig. 2b), and the one that had TUT selective responses "before" hearing LIVE TUT (Playback

1) in the Opto condition to see the effect of LC-NCM inputs on those neurons (Fig. 3d). These points are now more clearly explained in the text and figure legends.

- Authors should include the raw data points (response strengths for individual neurons) for all the bar plots showing mean response strength. For example, Fig 1g, 2d, 2f,2h, Extended data 1f, Extended data 2C, Extended data 3a,b - Raw data points (number of neurons) should be shown overlapping with the bar plots.
Thank you. Now we have included the individual data points overlaid on the average in all corresponding figures.

- Authors should include a summary table with information about firing rates for all the neurons during playbacks and live-tutor song experience to address some visualization inconsistencies such as between Line 74-75 and Extended Data Fig. 1a - For some of the trials, fast-spiking neurons seem to decrease their response to song playback as compared to their spiking before the song playback, such as for CON1 - Trial # 4-6, HET- Trial # 5, CON2 - Trial#4.

We now provide a summary table for the firing rate of all the neurons during playbacks and LIVE TUT as Supplementary Table 1

- Line 120-123 - In this section, authors should also mention the actual number of neurons in addition to the percentage of neurons.

We now provide the number of neurons in the text.

- Readers who might be working with other animal models such as rodents/primates may consider it an issue that there is a seemingly low number of neurons in the data for some figures to draw the conclusions. Early in the main text, authors should mention the total number of neurons recorded along with the fractions of silent and song-selective neurons for each figure. Authors may also acquaint the readers with the practical difficulties in performing single-unit recordings from LC and NCM in zebra finches. Authors may clarify to the readers that such numbers of neurons as recorded in this study are usual with zebra finches, if that is the case. For example, authors may provide a table, summarizing data from all the previous studies that have recorded single neurons with zebra finches. In the table, authors can include the brain regions (LC, NCM, and other brain regions) in one column, corresponding numbers of single units recorded from zebra finches in the second column, and relevant references in the third column.

Thank you for the valuable suggestions. We have now stated in the text the total number of neurons recorded from LC/NCM (163) and provided a table, and also how many were from LC or NCM, or responsive, or selective etc. in each related part. We now also describe that those numbers are comparable to those from studies using freely moving juvenile songbirds, while smaller than those in studies of other animals or in the anesthetized condition (164-). We believe that these comparisons will make the readers appreciate the practical difficulties of these experiments

- Line 181-182 and Extended Data Fig. 5b - For the left panel, the range of mean entropy is narrower for the opto-inhibited group, indicating lesser variability in the syllables produced by the opto-inhibited group as compared to the controls. Even though the range of syllables produced by the opto-inhibited group is similar to the birds isolated soon after hatching, it does not exclude the possibility of a loss of motor

plasticity from LC inactivation. Authors should provide additional references to justify that LC inhibition does not lead to major motor deficits.

Thank you for the suggestion. As you point out, we found that the range of mean entropy value for song syllables was narrower for the opto-inhibited group in adults. We suggested that was the effect of poor coping of tutor songs, but agree that we cannot exclude the possibility of opto-inhibition effect of motor deficits. We performed an additional analysis on Tet, Cackles and Stacks calls and found that the range of mean entropy value of calls were not different between opto-inhibited and control groups, suggesting opto-inhibition did not lead to major motor deficits. These data are now described in the text (1209-) and figures (Extended Data Fig. 6).

- Extended Data Fig 4. Not sufficient data to make the claims associated with fig 4. Authors may mention these findings in the discussion as the statistical power of these data seems insufficient to present them along with main results, unless they have some additional data to make these points.

We agree that the data is too small to draw conclusions, so we have removed those data from the manuscript.

- Line 557 - Authors should clarify how RS was estimated, preferably with a formula. The following is not clear to the reader: what were the durations of sound stimulus and pre-stimulus baseline period which were used to calculate the mean firing rate for estimating response strength

Response strength (RS) = $FR_{stim} - FR_{base}$, where FR_{stim} is the firing rate during the song presentation period (2s) and FR_{base} is the firing rate the same duration of period (2s) just before the song presentation.

This explanation is now described in the methods section with the formulas (1661).

- Line 522 - Reference 39 seems to be missing from the list. Authors should expand SAP2011 and provide a brief summary of how the song similarity was calculated using SAP 2011

Thank you for pointing this out. It was supposed to be reference 31 (now became 32), referring to Tchernichovski, O. et al. A procedure for an automated measurement of song similarity. Anim. Behav. 59, 1167–1176 (2000).

We also added the summary of how we measured song similarity using SAP 2011 in the Methods (1604-).

Minor concerns:

- Figure 1c. Red bars appear overlapping with non-uniform widths. Bar-width or x-axis can be readjusted to improve clarity in the figure.

Thank you. We corrected this and the revised panel is in the Extended Fig 1c.

- Fig 1d - It is an intriguing observation that TUT selective LC neurons stop their firing for 2.5 seconds when a tutor is introduced in the cage. The authors may further discuss the possible causes and effects of the cessation of firing in TUT-selective LC neurons. Do all the TUT-selective LC neurons stop firing reliably for 2.5 sec when a tutor is introduced? It would be interesting to check how the activity of NCM neurons varies around tutor introduction in opto-inhibition experiments (Fig 2) as compared to their activity in Fig 1.

Thank you for the suggestions. None of the LC neurons showed selective responses to any song stimuli, even after the presentation of LIVE TUT, which is now shown in Fig. 1d and f. Rather a pause of firing upon the tutor introduction occurred only in the FS LC neurons as shown in

Extended Data Fig. 1d. We found none of the NCM BS neurons paused their firing while it was difficult to find a pause as NCM BS showed lower spontaneous firing rates. We did not give Opto-inhibition when introducing a tutor in a cage in order to avoid unexpected effects on the upcoming response to LIVE TUT singing. Accordingly, we now discuss the neuronal circuits in the LC and NCM in the Discussion (1233-261)

- Figure 1f. 'dph' should be expanded in the figure legend
We have done this. (1289)

- Line 67: The sentence should include the information about the 30 min gaps between Playback 1-Live-tutor-payback 2. (For example, "...minutes of passive playback (Playback 2) at 30 min intervals each".)
Thank you. Now they are included (168-).

- Line 67: Recent evidence in rodents - 's' is missing from rodents.
Corrected.

- Line 121 - The sentence could be restructured to improve readability (For example, "However, a subset of otherwise silent NS neurons was activated only when...")
Thank you, we rephrased this as "While the number is small, a subset of otherwise silent NS neurons (n=9) was activated only when LC terminals were optogenetically inhibited..." (1178).

- Methods: Information about the light-dark cycle for housing the subjects should be added.
Now this information is added (1540).

- Line 449 - Authors should add a reference for the dox dosage.
We have now replaced this line with the data in which we used more conventional viral methods, and the data is shown in Fig. 3b.

- Line 476 - Stereotaxic coordinates for virus injection should be added to the methods.
We added this information in the Methods (1561).

- Line 488 - How many birds were implanted with a 16-ch silicon probe?
Five birds, which is now stated in line 571.

- Line 494-495 - Is this data presented in the results?
The data is now presented in Ext Data Fig.3.

- Line 504 - What was the approximate intensity of the 3s laser pulse?
Ten to 15 mW. This information is now included in the Methods (1588).

- Fig 3a - bar plots should be added
Bar plots with all individual data from BS neurons are now presented in Fig. 3c, and excluded the data for all NS neurons and only show the data from silent NS neurons in Extended Data Fig. 5b as suggested by another reviewer.

Reviewer #3 (Remarks to the Author):

In this manuscript, the authors use optogenetic methods show that projections for the Locus Coeruleus (LC) to the song nucleus NCM plays an important role in the social aspects of song learning. Neurons in NCM and LC respond more vigorously to song when presented by a live tutor vs. passive playback, and the response to passive tutor song showed a long-lasting increase after the tutoring sessions. However, when the projections from LC to NCM are optogenetically inhibited during the period of exposure to a live tutor, the increase in NCM activity appears to reverse. Furthermore, birds subject to optogenetic suppression during live tutoring make poorer copies of their tutor song than controls.

These results hold great promise for having a long-term positive impact on our attempts to understand the function of the LC. A key question for the field is the behavioral specificity of different LC afferents and efferents. The experimental control afforded by the songbird learning paradigm - both sensorimotor and social - may provide a powerful paradigm for investigating these issues.

However, the paper suffers from interlocking issues related to the complexity of the results and the clarity with which they are presented. Furthermore, there may be a question about whether the songs used for stimuli (both tutoring and control) are appropriately counterbalanced.

Thank you for reviewing our manuscript. We are glad that the reviewer sees the promise in our results for understanding LC function underpinning behavior and we appreciate your valuable comments. We provide our answer to each individual comment below.

It was difficult to track all the separate experiments. In figure 2h. Response strength (RS=response-background) to Tutor, conspecific 1 and conspecific 2 and heterospecific song are shown before and after exposure to the live tutor with opto suppression of LC inputs. The conclusion from the figure is that RS goes down selectively for the tutor. But during playback 1 TUT, CON1 and CON2 are just three different zebra finch songs the juvenile has never heard before. But response strength varies quite a bit between songs, especially between TUT and CON2. If the initial response was like CON2 would you be able to obtain this result? This is a common problem in the song field in that all randomly selected songs have differences that are consistent across birds and neurons. Dealing with these issues relies on careful experimental design and counterbalancing. The authors may have enough data to rule out this as a possibility, but readers need to be able to track this down to be confident of the results.

Thank you for your suggestions. We allowed tutor exposure for four days for each juvenile (and three days for Opto-condition), and on each day they were provided the song playback before (Playback 1) and after (Playback 2) the live tutor song exposure (LIVE TUT). We used the data from a different group of neurons for the previous Fig 1h and Fig 2h. Fig 1h represented the BS neurons which acquired TUT selective responses “after” hearing LIVE TUT singing (Playback 2), while Fig. 2h were for the neurons which showed TUT selective responses “before” hearing LIVE TUT (Playback 1). That is why the average RS for Playback 1 was higher in Fig. 2h. We now show the average RS to TUT in all BS neurons with all the data points from each neuron in the normal (Fig. 2a) and Opto- (Fig. 3c) condition, which show no difference in their distribution. In addition, we again provided the data from the BS neurons which acquired TUT selective responses “after” hearing LIVE TUT singing (Playback 2) in the normal condition (Fig. 2b), and the one which had TUT selective responses “before” hearing LIVE TUT (Playback 1) in the Opto condition to see the effect of LC-NCM inputs on those neurons (Fig.

3d). We further found that there was no increase in the number of TUT selective neurons after LIVE TUT with opto-inhibition (Fig. 3g), which was a big contrast to the result in the normal condition (Fig 2e). We believe all of these points are now clearly explained in the text.

A second major issue is the complexity of the results. The authors show intriguing differences in responses between different neuron classes defined by spike width and spontaneous activity. However, these differences are not laid out clearly. The section between line 79 and 101 is particularly dense and describes an entire series of results with little commentary. But not a single one of these response type differences was important enough to mention in the discussion. The authors should give serious consideration to paring the paper down to the main results and discussing response-type differences in a more extensive paper. If they do want to retain cell-type differences, it may be useful to present the main results so that the reader can follow them through, and then show how these properties are distributed across different cell-types. We did that now.

We are currently more focused on the BS neurons in the NCM, as we found that a subset of BS neurons acquired TUT selective responses by exposure to LIVE TUT and Opto-inhibition of LC inputs disrupted the increase of TUT selective neurons. While we noted a small subset of silent NS neurons that responded only to laser stimulation (Extended Data Fig. 5), and showg some data of NS neurons in the Extended Data Fig. 2e-g. We added further discussion of the possible neuronal circuits in the LC and NCM in the Discussion that includes coverage of these points (1231-261).

There are many other minor issues mostly having to do with clarity. I will cover a representative sample. The authors state that they “characterize a neural circuit for social authentication” (14). The results only investigate the projection from LC to NCM - at best one component of the social authentication circuit. The authors show a hint of a projection from the thalamus to LC, but this is in supplemental data. Is LC the place where “social authentication is computed”, or does LC broadcast the authentication signal to modulate plasticity?

The authors distinguish their results from effects on song prosody. Issues related to timing and sequencing have been tied to song nucleus HVC and previous results show that social interaction modulates dopamine-dependent mechanisms related to HVC. Readers need to be given better explanation to understand distinctions between different aspects of song learning that are only clear to experts.

We revised all of the text, especially in the results sections, which, we believe, should improve the clarity of these points.

There were many phrases about “reconfiguring responses”, “disordered responses,” “modulation of activity” and effects on song learning. Examples include

- 18. “LC firing reconfigured long-term song-selective neural responsiveness”

We rephrased this to “LC activity regulated” (117).

- 20. “optogenetic inhibition of LC presynaptic signaling in the NCM disordered NCM neuronal responsiveness to live tutor singing”

We revised this sentence to read “optogenetic inhibition of LC presynaptic signaling in the NCM reduced NCM neuronal responsiveness to live tutor singing and impaired song learning”

- These are two section headings. Do they mean something different?

- o 103. LC inputs configure auditory responses of NCM neurons
- o 136 Live tutor singing modulates NCM neuron activity via LC inputs

To fix this we merged these two sections.

- 157. “reorganize and maintain neuronal circuit activity in the NCM via functional inputs from the LC.” While the authors have shown that optogenetically inhibiting LC-NCM projections disrupts learning, it is not clear whether LC is affecting activity that then alters learning, or modulates plasticity/gene expressions that effects how or how strongly NCM activity patterns are written into long-term changes, or something in between or something else. The authors should be careful about statements like LC inputs reconfigure LCM activity since this can mean many things.

This sentence is now revised to read “vocal communication with a tutor is essential to modulate neuronal circuit activity in the NCM via functional inputs from the LC to acquire song-selective auditory responsiveness in the LC-NCM neural circuit” (1184-). We describe that a subset of NCM neurons acquire selective auditory responses to TUT by exposure to LIVE TUT singing. Opto-inhibition of LC-NCM activity during LIVE TUT singing prevents NCM neurons from developing TUT selective response in the short-term (hours), and impairs juveniles from learning songs from a tutor in the long-term (weeks). But, as the reviewer pointed out, how the neuronal activity changes in the NCM in the short-term lead to effects on song learning in the long term remains unknown. We discussed several possible mechanisms in the Discussion.

51. “We observed enhanced neural activity in the LC-NCM neuronal circuit during vocal communication with a live adult tutor. Optogenetic inhibition of the LC-NCM neural circuit during exposure to a live singing tutor reduced song learning.” The authors have shown some optogenetic reduction in activity, but it is not clear that the reduction in activity during tutor song singing is the main cause of the reduced song learning. *Yes, we did not show how reduced activity during LIVE TUT is a direct cause of less song learning. Rather, we show that Opto-inhibition of LC-NCM circuit disrupted developing TUT-selective response in NCM neurons and song learning. We have now rephrased the sentence to read “Juveniles, in which the LC-NCM neural circuit was optogenetically inhibited during exposure to a live singing tutor, did not learn the tutor's song r” (151), and we discuss possible mechanisms in the Discussion (1231-261).*

194. “The noradrenergic LC provides an unexpected synaptic perspective on the learning of social communication.” 208. “Synaptic mechanisms for NE signaling on short-term plasticity in postsynaptic cortical neuron (23) or LTP (24, 25) are known, but synaptic changes correlated to long-term physiological and behavioral outcomes, as we show in NCM, are rare.” I’m not in this field, but are the authors’ results really unexpected? Is it rare to show LC having long term effects on behavior? If so, please discuss further. *We have rephrased these sentences to read “Synaptic mechanisms for NE signaling on short-term plasticity in postsynaptic cortical neurons (24) or LTP (25, 26) are well known, and LC-derived NE/DA signaling in the hippocampus has been suggested to enhances memory consolidation through LTP. Here we showed that direct inhibition of LC activity during hearing of the LIVE TUT prevents song learning from the tutor, and further suggesting an LC contribution to long-term memory formation.” (1244)*

222. “DA and NE signaling working in parallel to continuously monitor the quality of social learning.” DA and NE signaling do not monitor the quality of social learning. There are several places in the paper where the authors could take greater care in their wording.

Thank you for pointing this out. We revised the text and believe there is no further confusion.

225. “Our results in songbirds annotate this emerging framework, adding the authentication of social interactions to brain-wide networks.” Not sure what this means.

The authors mention that juveniles can learn if they actively trigger playback. This would be a great experiment in the current context!

We now rephrased the sentence to read “Our results in songbirds annotate this emerging framework, by exploring a specific neuronal circuit for authenticating social information in vocal learning” (1274). We agree that the suggested experiment is great as a next step.

Figures:

It seems that the TUT song RS for playback 1 is generally quite low (<1) in 1h, 2d and 2f, but then TUT song RS is about 5 in 2h. Also, from, 2d and 2f it seems that RS stays the same or increases from playback 1 to 2 with opto, but in 2h it decreases. What is going on here?

We are sorry that the data presentation was confusing. As we also noted to the other reviewers, we used the data from a different subset of neurons for the previous Fig1h and Fig2h. Fig1h represented the BS neurons which acquired TUT selective responses “after” hearing LIVE TUT singing (Playback 2), while Fig. 2h was for neurons that showed TUT selective responses “before” hearing LIVE TUT (Playback 1). That is why the average RS to Playback 1 was higher in Fig. 2h.

We now show the average RS to TUT in all BS neurons with all the data points from each neuron in the normal (Fig. 2a) and Opto- (Fig. 3c) conditions, which show no difference in their distribution. In addition, we again provided the data from the BS neurons that acquired TUT selective responses “after” hearing LIVE TUT singing (Playback 2) in normal condition (Fig. 2b), and the one which had TUT selective responses “before” hearing LIVE TUT (Playback 1) in the Opto condition to see the effects of LC-NCM inputs on those neurons, respectively (Fig. 3d). These points are now more clearly explained in the text and figure legends.

I don't understand 2f. I think the legend is wrong. But if I'm reading it right, the total response may stay about the same or go down, but RS goes up since baseline firing goes down? This needs explanation.

We found that the RS (Firing rate during song stim – Firing rate during no sound stim (baseline)) significantly decreased after hearing LIVE TUT with Opto-inhibition, while the baseline FR did not change. This data is now presented in Fig 3d and the baseline firing rates are described in the text (1160).

It seems that all of figure 3 and figure 2c-f are making the distinction between NS and RS NCM neurons. Seems like a lot of space given that little is made of the distinction. In 2c-f I would suggest NS and RS be somewhere in the subplot title or y axis label so that a reader doesn't have to dig through the legend to see what distinctions are being made (like in fig 3). It's hard to trace all the bars in 3a. Perhaps a scatter plot with playback 1 along the x axis and playback 2 along the y axis plus a dashed diagonal? That might even make it easier to convey the information in 3b (perhaps eliminate it)? It seems that 3c is kind of redundant with fig 2 and 3d is similar to 3b. What is the bottom line here? Lot's of graphs, hard to figure out what to make of them.

We apologize for the confusion. As we also answered to other reviewers, we are currently focused more on the BS neurons because we found that a subset of BS neurons acquired TUT selective responses after hearing LIVE TUT, and Opto-inhibition prevented the increase in the

proportion of TUT-selective BS neurons. We labeled the BS (or NS) neurons in the figures and removed redundancies. We hope the reviewer will find the revised manuscript more readable.

Figures 1 and 4 are well done.

REVIEWER COMMENTS

Reviewer #1 (Remarks to the Author):

The authors have done a thorough job in their revision. My concerns have been adequately addressed.

Reviewer #2 (Remarks to the Author):

The authors have done a commendable job in revising the manuscript by addressing all the reviewer comments. The manuscript is more clear as the information flows well from beginning to end. Overall, I really enjoyed the range of empirical investigations from behavior, neural activity manipulation, anatomical confirmations, and song similarity analyses. I have a few minor concerns listed below and addressing them can further clarify the findings for the readers.

- All the data for playback 1 and 2 ('Playback' factor for 2-way ANOVA) for TUT, CON1, CON2, HET ('Auditory stimulus' factor for 2-way ANOVA) should be analyzed with two-way repeated-measures ANOVA to gain clear insight into the possible statistical interactions across different conditions. These observations could be discussed in the manuscript in the appropriate section.

- Line 100-102 - What is the inference from this observation?

- For every first usage of a term, an expanded version should be used with an abbreviation in the parenthesis next to it. E.g. Line 106 - broad spiking should be abbreviated (BS) in parenthesis next to it. Line 128 NS should be expanded and abbreviation should be placed in parenthesis next to the expanded words.

- RS has two meanings in the manuscript: Regular spiking LC neurons and response strength. Even though authors have avoided using RS for regular spiking neurons, due to other abbreviations such as FS, BS, and NS, the reader automatically imagines RS for regular spiking and it gets confusing when response strength is abbreviated as RS. Authors should clearly demarcate the difference in their abbreviations (for example 'RS' for Regular spiking, and 'rs or RSt' for response strength)

- Line 150-151- How was optogenetic inhibition restricted to the period when Juveniles were hearing LIVE TUT but not when they were merely interacting with the tutor? Was there a sound-controlled

feedback loop to turn on/off the laser impulses that were activated by LIVE TUT? The temporal delays in stimulation and corresponding neural inhibition need to be considered together and explained if such a feedback loop was implemented.

- The methods section should have an independent subsection for optogenetics and the details to clarify the aforementioned concern should be added to it.

- Line 165-166 -Ext data Fig 3c is a good control for optogenetics experiments but the mean RS to TUT for all-BS-NCM neurons looks much higher, as compared to the mean-RS for TUT-selective BS-NCM neurons (Figure 2g and ext data Fig 2b). Although data in Ext Fig 3c provides sufficient evidence for control optogenetics experiments, it would be better to add the data for mean RS to TUT for TUT selective neurons in the control animals where the laser was shined at LC projections in NCM with only GFP expression.

- Line 177-78 - Extended data fig 4a can be moved to the main 'fig 4' as it provides a clear picture of neuron numbers involved in different conditions and having to go to 'extended fig 4a' breaks the flow of the reader.

- Line 178-180 - The sentence “While the number....Ext data Fig.5a-b” should be moved to the end of line 183 as it breaks the flow of the reader.

- Line 194 - 195 - cite reference #32 for song similarity somewhere near the first mention of song similarity.

- Future directions - For future experiments, it would be nice to identify the aspects/features of tutor presence that are used by juveniles for social authentication such as identifying the contribution of odor and visual cues in social authentication. Bird vision is highly developed and is it possible that the juveniles look at the beak or maybe even vocal cord movements for social authentication, or it could be the tutors' smell. Interesting experiments such as having a dummy tutor with TUT playback can bring deeper insights into the specific features of social interactions responsible for social authentication.

Reviewer #3 (Remarks to the Author):

The revised manuscript is significantly improved. The main flow of the results as conveyed by the figures is clear and impactful. The discussion is also well organized and clear. Most issues regarding complication in the main text have been largely addressed, and the authors do a good job of not getting bogged down in the details. The presentation of cell-type specific results is also handled well.

However, there are several remaining issues related to the clarity of presentation. These should be able to be addressed with some modest revision of the text.

1. The language related to figure 1 does a poor job of conveying the main point: after exposure to the LIVE TUT on the first day, LC neurons show greater response to passive playback of ALL songs. The interesting result here is that increases are NOT specific to the TUT. But the title for figure 1 is "LC neurons increase auditory responsiveness to tutor song playback after social interaction with a singing tutor," which gives the impression that the increase is specific to TUT. Also, in the text the increased response is described as applying to TUT "as well as" other songs. Then the authors describe some details, and then mention the lack of tutor song selectivity. The main point gets completely buried in the passive language.

2. The authors state "We found a small number of neurons (n=4) that increased their RS to TUT playback and began to show TUT-selective responses on the second day of tutoring, but did not find those neurons later than the third day of tutoring (Fig. 2f-g)." The wording here is confusing. The authors did see TUT-selective neurons after the third day of tutoring. What they did not find is any further increases in selectivity.

3. The authors are inconsistent in their use of the term prosody. In human speech prosody refers to rhythm and stress and is contrasted with and distinct from phonetic cues. In the Discussion the authors contrast the authentication of social cues vs. the processing of "high-resolution prosodic vocal information, like the auditory cortex." The auditory cortex processing BOTH prosodic vocal information as well as featural (phonetic) aspects. Sometimes the authors seem to use prosody interchangeably with high-resolution acoustic features ("song's prosodic features for accurate song memory"; "song prosody, namely syllable acoustical structures") and sometimes prosody is presented as separate aspect of song acoustics ("prosodic or acoustic features of the song"). None of the data here distinguishes prosodic from other specific features of song acoustics. The presentation would be much clearer if the authors used more general language (like "acoustic features" or "song specific acoustics") to contrast with the social cues and drop the use of the term prosody.

4. I do not understand this sentence: "In contrast LC neurons showed greater responses to LIVE TUT later than the third day of tutoring, suggesting that plasticity would occur in a specific subset of postsynaptic (NCM) neurons which might express specific NE receptors." The phrase "plasticity would occur" is particularly confusing. In general, I don't see how LC responses can be attributed to particular "(NCM) neurons which might express specific NE receptors."

Reviewer #1 (Remarks to the Author):

The authors have done a thorough job in their revision. My concerns have been adequately addressed.

Thank you for reviewing our manuscript. We are glad to know the reviewer satisfied with our job.

Reviewer #2 (Remarks to the Author):

The authors have done a commendable job in revising the manuscript by addressing all the reviewer comments. The manuscript is more clear as the information flows well from beginning to end. Overall, I really enjoyed the range of empirical investigations from behavior, neural activity manipulation, anatomical confirmations, and song similarity analyses. I have a few minor concerns listed below and addressing them can further clarify the findings for the readers.

Thank you for reviewing our manuscript. We are glad to know the reviewer enjoyed reading our revised manuscript. We further revised our manuscript according the reviewer's suggestion and provide answers for one-to-one points below.

- All the data for playback 1 and 2 ('Playback' factor for 2-way ANOVA) for TUT, CON1, CON2, HET ('Auditory stimulus' factor for 2-way ANOVA) should be analyzed with two-way repeated-measures ANOVA to gain clear insight into the possible statistical interactions across different conditions. These observations could be discussed in the manuscript in the appropriate section.

We conducted two-way ANOVA statistics for the comparisons between neuronal response between Playback 1 and 2 with multiple song stimuli. We provide the results in relevant places within text (e.g. 1104).

- Line 100-102 - What is the inference from this observation?

LC neurons kept showing greater responses to LIVE TUT over the days of tutoring which contrasted with the response of NCM TUT-selective neurons which stopped showing greater responses to LIVE TUT after three days of tutoring. Those were important to discuss about the possible neuronal circuit and plasticity mechanism for song learning together with other results. We revised the sentences in the discussion and made the link with this result clear (1271-).

- For every first usage of a term, an expanded version should be used with an abbreviation in the parenthesis next to it. E.g. Line 106 - broad spiking should be abbreviated (BS) in parenthesis next to it. Line 128 NS should be expanded and abbreviation should be placed in parenthesis next to the expanded words.

We corrected.

- RS has two meanings in the manuscript: Regular spiking LC neurons and response strength. Even though authors have avoided using RS for regular spiking neurons, due to other abbreviations such as FS, BS, and NS, the reader automatically imagines RS for regular spiking and it gets confusing when response strength is abbreviated as RS. Authors should clearly demarcate the difference in their abbreviations (for example 'RS' for Regular spiking, and 'rs or RSt' for response strength)

Thank you for the suggestions. Now Response strength is abbreviated as “RST” and revised the texts and figures accordingly.

- Line 150-151- How was optogenetic inhibition restricted to the period when Juveniles were hearing LIVE TUT but not when they were merely interacting with the tutor? Was there a sound-controlled feedback loop to turn on/off the laser impulses that were activated by LIVE TUT? The temporal delays in stimulation and corresponding neural inhibition need to be considered together and explained if such a feedback loop was implemented.

- The methods section should have an independent subsection for optogenetics and the details to clarify the aforementioned concern should be added to it.

We applied the three sec-long laser pulse manually each time a tutor sung. In case a tutor song lasted more than three seconds, we applied another 3 sec-long laser pulse until a tutor stopped singing.

Now we provide detailed methods for optogenetic inhibition experiment in an independent section in Methods including above information (1605~).

- Line 165-166 -Ext data Fig 3c is a good control for optogenetics experiments but the mean RS to TUT for all-BS-NCM neurons looks much higher, as compared to the mean-RS for TUT-selective BS-NCM neurons (Figure 2g and ext data Fig 2b). Although data in Ext Fig 3c provides sufficient evidence for control optogenetics experiments, it would be better to add the data for mean RS to TUT for TUT selective neurons in the control animals where the laser was shined at LC projections in NCM with only GFP expression.

*We found a small subset of BS neurons which showed higher firing rates. Now we provide the data for mean RST to TUT of TUT-selective neurons (including both the neurons which were selective at Playback 1 already and kept it and the others which acquired TUT selectivity by hearing LIVE TUT singing as **Extended Fig 3c (right)**).*

- Line 177-78 - Extended data fig 4a can be moved to the main 'fig 4' as it provides a clear picture of neuron numbers involved in different conditions and having to go to 'extended fig 4a' breaks the flow of the reader.

We moved it according to the suggestion.

- Line 178-180 - The sentence “While the number....Ext data Fig.5a-b” should be moved to the end of line 183 as it breaks the flow of the reader.

We changed the order as suggested.

- Line 194 - 195 - cite reference #32 for song similarity somewhere near the first mention of song similarity.

We cite the paper in the main text now as Ref #20 (1210).

- Future directions - For future experiments, it would be nice to identify the aspects/features of tutor presence that are used by juveniles for social authentication such as identifying the contribution of odor and visual cues in social authentication. Bird vision is highly developed and is it possible that the juveniles look at the beak or maybe even vocal cord movements for social authentication, or it could be the tutors' smell. Interesting experiments such as having a dummy tutor with TUT playback can bring deeper insights into the specific features of social interactions responsible for social authentication.

Thank you for the suggestions!

Reviewer #3 (Remarks to the Author):

The revised manuscript is significantly improved. The main flow of the results as conveyed by the figures is clear and impactful. The discussion is also well organized and clear. Most issues regarding complication in the main text have been largely addressed, and the authors do a good job of not getting bogged down in the details. The presentation of cell-type specific results is also handled well.

However, there are several remaining issues related to the clarity of presentation. These should be able to be addressed with some modest revision of the text.

Thank you for reviewing our manuscript. We are happy to learn that the reviewer found our revised manuscript was improved. We further revised our manuscript according to the suggestions from the reviewer.

1. The language related to figure 1 does a poor job of conveying the main point: after exposure to the LIVE TUT on the first day, LC neurons show greater response to passive playback of ALL songs. The interesting result here is that increases are NOT specific to the TUT. But the title for figure 1 is "LC neurons increase auditory responsiveness to tutor song playback after social interaction with a singing tutor," which gives the impression that the increase is specific to TUT. Also, in the text the increased response is described as applying to TUT "as well as" other songs. Then the authors describe some details, and then mention the lack of tutor song selectivity. The main point gets completely buried in the passive language.

Thank you for the suggestion. We agree that the point there is LC increased their response strength to the playback of ALL songs after hearing LIVE TUT singing. We revised the text (197) and figure legends (1306) to make this point clear.

2. The authors state "We found a small number of neurons (n=4) that increased their RS to TUT playback and began to show TUT-selective responses on the second day of tutoring, but did not find those neurons later than the third day of tutoring (Fig. 2f-g)." The wording here is confusing. The authors did see TUT-selective neurons after the third day of tutoring. What they did not find is any further increases in selectivity.

We found the TUT selective neurons at Playback 1 (before hearing LIVE TUT singing) but we did not that those neurons increase response strength to TUT after LIVE TUT after third day of tutoring. We did not see increase of the number of TUT selective neurons after hearing of LIVE TUT after the third day of tutoring either. We revised the text as "We found a small number of neurons (n=4) that increased their RST to TUT playback and began to show TUT-selective responses on the second day of tutoring. But later than the third day of tutoring, we did not find the neurons which increased RST or selectivity to TUT playback after hearing the LIVE TUT (Fig. 2f-g). (1125)".

3. The authors are inconsistent in their use of the term prosody. In human speech prosody refers to rhythm and stress and is contrasted with and distinct from phonetic cues. In the Discussion the authors contrast the authentication of social cues vs. the processing of

"high-resolution prosodic vocal information, like the auditory cortex." The auditory cortex processing BOTH prosodic vocal information as well as featural (phonetic) aspects. Sometimes the authors seem to use prosody interchangeably with high-resolution acoustic features ("song's prosodic features for accurate song memory"; "song prosody, namely syllable acoustical structures") and sometimes prosody is presented as separate aspect of song acoustics ("prosodic or acoustic features of the song"). None of the data here distinguishes prosodic from other specific features of song acoustics. The presentation would be much clearer if the authors used more general language (like "acoustic features" or "song specific acoustics") to contrast with the social cues and drop the use of the term prosody.

Thank you for the suggestions. We agree that it was confusing and here we contrast the acoustical features of songs and social cues of tutor singing. We removed the word "prosody" and replaced as "acoustic features" of songs in the current manuscript.

4. I do not understand this sentence: "In contrast LC neurons showed greater responses to LIVE TUT later than the third day of tutoring, suggesting that plasticity would occur in a specific subset of postsynaptic (NCM) neurons which might express specific NE receptors." The phrase "plasticity would occur" is particularly confusing. In general, I don't see how LC responses can be attributed to particular "(NCM) neurons which might express specific NE receptors."

Here we found LC neurons kept showing greater responses to LIVE TUT after the second day of tutoring, while none of LC neurons acquired TUT selectivity. In contrast, in the NCM a subset of BS neurons acquired TUT selective responses, and those neurons did not show greater response to LIVE TUT, neither did increase RST to TUT after LIVE tutor after the third day of tutoring. Increase of RST to TUT did not occur even at the first day of tutoring when LC axonal activities in the NCM was optogenetically inhibited. Those suggests LIVE TUT singing triggers higher LC neuronal activities and NE release from LC neurons, that might cause synaptic plasticity to acquire greater responses to TUT in a subset of NCM neurons which expresses NE receptors.

We revised the texts to make those clear (1270~).

REVIEWERS' COMMENTS

Reviewer #2 (Remarks to the Author):

All my concerns have been well-addressed.

Reviewer #3 (Remarks to the Author):

The authors have adequately addressed all of my concerns.